



# Evidence of Changes in Sedimentation Rate and Sediment Fabric in a Low Oxygen Setting: Santa Monica Basin, CA

Nathaniel Kemnitz[1], William Berelson[1], Douglas Hammond[1], Laura Morine[1], Maria Figueroa[3], Timothy W. Lyons[3], Simon Scharf[1], Nick Rollins[1], Elizabeth Petsios[1], Sydnie Lemieux[2], and Tina Treude[2]

[1]University of Southern California, Los Angeles, California, U.S.A
[2]University of California, Los Angeles, California, U.S.A
[3]University of California, Riverside, California, U.S.A

Correspondence to: Nathaniel Kemnitz (kemnitz@usc.edu)

**Abstract.** The Southern California Bight is adjacent to one of the world's largest urban areas, Los Angeles. As a consequence, anthropogenic impacts could disrupt local marine ecosystems due to municipal and industrial waste, pollution, and flood control measures. Superimposed on the growth of an urban metropolis, the impact of climate change has been felt most strongly over the past 50 years in terms of rising $pCO_2$ and warming. Santa Monica Basin (SMB), due to its unique setting in low oxygen and high sedimentation environment, has provided an excellent sedimentary paleorecord of these anthropogenic changes. This study examined ten sediment cores, collected from different parts of the SMB between spring and summer 2016, and compared them to existing cores in order to document changes in sedimentary dynamics during the last 250 years, with an emphasis on the last 40 years.

Mass accumulation rates (MAR) for the deepest and lowest oxygen-containing parts of the SMB basin (900-910m) established using $^{210}Pb$ have been remarkably consistent during the past century, averaging $17.5 \pm 2.1$ mg/cm$^2$-yr. At slightly shallower sites (870-900m), accumulation rates showed more variation, but yield the same accumulation rate, $17.5 \pm 5.5$ mg/cm$^2$-yr. Excess $^{210}Pb$ sedimentation rates were consistent with rates established using bomb-test $^{137}Cs$ profiles. However, $^{14}C$ profiles from cores collected in the deepest part of the SMB, where fine laminations are present up to 250 years B.P., indicate that MAR was slower prior to ~ 1900 CE (rates obtained = 9 and 12 mg/cm$^2$-yr). $\delta^{13}C_{org}$ profiles show a relatively constant value down core suggesting that the change in sediment accumulation rate is not accompanied by a change in organic carbon sources to the basin. The increase in sedimentation rate towards the recent occurs at about the time previous studies predicted an increase in siltation and the demise of a shelly shelf benthic fauna on the SMB shelf.

X-radiographs show finely laminated sediments in the deepest part of the basin only, with cm-scale layering of sediments or no layering whatsoever in shallower parts of the SMB basin. The absence of finely laminated sediments in MUC 10 (893 m) and MUC 3 (777 m) suggest that the rate at which anoxia is spreading, has not increased appreciably since cores were last analyzed in the 1980s. Based on core top data collected during the past half century, sedimentary dynamics within SMB has changed minimally during last 40 years. Specifically, mass accumulation rates, laminated sediment fabric, extent of



bioturbation, and % $C_{org}$ have not changed.  The only parameter that appeared to have changed in the last 250 years was the MAR with an apparent step-wise increase occurring between ~1850-1900 CE, yet the post-1900 CE constancy of sedimentation through a period of massive urbanization is surprising.

## 1 Introduction


The use of laminated sediments as a record of environmental change has many historical precedents (Koivisto and Saarnisto, 1978; Gorsline, 1992; Algeo et al., 1994). The deepest portion of Santa Monica Basin (SMB, Fig. 1) has been accumulating finely laminated sediments for the past approximately 400 years (Christensen et al., 1993). The presence of fine lamination is evidence that macrofaunal activity on or in the sediment has been minimal to absent. Savrda et al. (1984) documented the

transition from laminated to bioturbated sediments against an oxygen gradient in the bottom water, which is the chief control of benthic macrofauna presence (Levin, 2003). Yet two things are necessary to produce laminated sediments.  First is the absence of disturbance or mixing, and the other is a pulsed delivery of sediment that produces a distinction in composition or sediment fabric (Kemp, 1996).

Sediment trap studies at a long-term study site (SPOT, Fig. 1) in San Pedro Basin, which is adjacent to SMB, demonstrated a seasonal pattern of sedimentation with highest rates in late winter and spring (Collins et al., 2011). Similarly, Haskell et al. (2015) documented seasonality in upwelling velocity and biogenic particle export from the upper ocean at SPOT. In contrast to the seasonal forcing of sedimentation in local waters, sediments in the SMB show primarily non-annual laminations with a frequency of 3-7 year. This lamination cycle may be consistent with the frequency of heavy rainfall during El Nino years, as

they typically result in higher than average Southern California rainfall (Quinn et al., 1978; Christensen et al., 1993)

The present study considers changes in SMB sedimentation over the past 150 years, a time period when changes in ocean biogeochemistry have been observed both globally and regionally. For example, the large-scale changes in the size and intensity of global oxygen minimum zones (Stramma et al., 2010; Brietburg et al., 2018) have also been documented for

Southern California waters (Bograd et al., 2008). Oxygen concentrations in near-surface waters of the Southern Californian shelf show a 20- to 50-year decline beginning in the early 1960s, which was attributed to increased stratification and/or increased productivity caused by enhanced nutrient supply (Booth et al., 2014). In concert with changes in upper ocean oxygen content, publications by Huh et al. (1989) and Christensen et al. (1993) have documented expansion of the area of laminated sediments in SMB. X-radiography showed that homogenized sediment was covered by laminated sediment, marking a

transition from bioturbation to lamination preservation.

Age dating of this transition, as deduced by recent $^{210}$Pb sedimentation rate, revealed concentric zones covering the entire basin floor, which accumulated laminated sediments from 400 (basin center) to 50 (shallower depths) years before present (ybp).





This expansion in lamination is taken as evidence of expanding oxygen deficiency and is particularly interesting given the
global and local changes mentioned above. Over the past 400 years of laminae accumulation in SMB, the southern California
region has grown into one the world's largest urban areas.  Particularly notable was a change in ecosystem structure of benthic
shelf fauna during the mid- to late 1800's, which was attributed to the onset of higher coastal sediment delivery caused by
grazing cattle.  This transition increased the frequency and amount of sediment entering the coastal zone (Tomašových and
Kidwell, 2017). Another notable anthropogenic impact is the introduction of sewage waste into the coastal system starting in
the early 1900's (Alexander and Venherm, 2003). Advanced treatment of this sewage did not start until the 1970's.
Furthermore, channelization of the LA River and construction of sediment-trapping flood basins up-river have occurred over
the past century. Thus, there is ample evidence of environmental change in and around the SMB over the past 150 years.

Starting with a study by Bruland et al. (1974), investigators have been using $^{210}$Pb profiles of sediments as a means of
documenting sediment accumulation and sediment mixing in the SMB. A compilation of core analyses was published by Huh
et al. (1989), and further work by Alexander and Lee (2009) provided a record of sedimentation in the SMB from the 1970's
through the 1990's. Our work here (conducted in 2016) aimed at augmenting this record of coastal sedimentation, quantified
by analyses of $^{210}$Pb and $^{14}$C profiles. We sampled intact surface sediments (top ~30 cm) from the SMB and also conducted
analyses of (1) sediment fabric by x-radiography, (2) sediment macrofaunal composition, and (3) $C_{org}$ content to study changes
in sedimentation in the SMB over the past 150 years. Our study provides new information about sedimentation and the potential
expansion or contraction of laminated sediments over the past 150 years with a focus on the past 40 years.

## 2 Methods

### 2.1 Study Area

The San Pedro-Santa Monica Basins are 'bathtub'-shaped and oriented north-west to south-east adjacent to the Los Angeles
coastline. The two basins, both approx. 900 m deep, are separated by a sill. Water entering San Pedro Basin (SPB) from the
south-east crosses the sill at ~740 m and then passes into the SMB (Hickey, 1991). Bottom water circulation below the sill
depth is sluggish, < 1.0 cm/s and generally moves in a counter-clockwise direction. To the north-east of SMB is a slope and
the broad Santa Monica shelf, which is incised by Redondo Canyon, in the south-east portion of the basin, and Santa Monica
Canyon, which empties in the middle of SMB; Malibu and Pt. Dume Canyons drain into the northeast portion of the basin.
Sedimentation to the SMB is characterized as hemi-pelagic, interrupted by sandy turbidites that primarily originate from the
northeast canyons and spread onto the basin floor (Gorsline, 1992).


The upper ocean waters (above 300 m) are a mixture of at least two distinctive water masses (Fig. 2) whereas the waters below
sill depth have a T-S signature suggestive of mixing with a water mass that originates somewhere in the NW Pacific (Lynn



and Simpson, 1987). All waters below 400 m are low in oxygen (<20 µM) although the deepest water sometimes has slightly

higher concentrations compared to those immediately above (Fig. 2). This phenomenon is rare and identifies a basin 'flushing'

event (Berelson, 1991). Generally, water enters SMB, and the sluggish circulation and slow rates of replenishment (deep water

residence times on the order of 1-3 years; Hammond et al., 1990) tend to deplete oxygen further. Hence oxygen concentrations

range between 1 - 9 µM (Berelson, 1985). Complete bottom water depletion of oxygen and/or the presence of sulfide in bottom

waters has never been reported. The sediments of SMB have 15-20 wt. % $CaCO_3$, 2-6 wt. % $C_{org}$ and 2-8 wt. % $SiO_2$ (Cheng

et al., 2008).


### 2.2 Water column and sediment sampling

Temperature, salinity, and dissolved $O_2$ concentrations were profiled in the water column of the SMB (0-907 m water depth)

from aboard the RV Yellowfin (Southern California Marine Institute) in April 2016, using a CTD (Sea-Bird 25) with attached

SBE43 oxygen sensor (calibrated by Winkler titration). For CTD calibration, automated bath systems, sensor stability, primary

standards in temperature (water triple point and gallium melting point) and conductivity (International Association for the

Physical Sciences of the Oceans: IAPSO) were maintained.

Ten sediment cores were collected in April and July 2016 from eight stations (MUC 3, MUC 5-11) between 319 and 907 m

water depth, using a miniature multicorer (MUC, K.U.M. Kiel) equipped with four polycarbonate core liners (length: 60 cm,

inner diameter: 9.5 cm).

After cores were retrieved, one core was sectioned shipboard in one-centimeter intervals in the upper 10 cm and two-centimeter

intervals below 10 cm. Aliquots were delivered to porosity vials and the remaining mud placed in plastic bags. A second core

from the same multicore deployment was preserved intact for x-radiography.


### 2.3 Porosity and Integrated Mass

Wet mud from the sectioned core was placed in pre-weighed porosity vials (15 mL snap-cap glass vials), re-weighed and dried

at 50°C for 48-96 hours. Vials were subsequently re-weighed to determine water loss. The dry weight was corrected for salt

content, assuming a salinity of 35. Porosity was determined assuming a grain density of 2.5 g/cm$^3$. Integrated mass to the mid-

point of each sample interval was calculated from the porosity, this density, and summing to numerically integrate eq. E-1:

$$I = \int (1 - \emptyset)\rho \Delta x \tag{1}$$

where $\Delta x$ is the interval thickness, $\rho$ is solid phase density, I is integrated mass, and $\emptyset$ is porosity.



### 2.4 Macrofauna

Sediment from one of the cores collected from each site was used for faunal surveys. The first 5 cm of each core was sectioned
into 1 cm intervals for the purposes of capturing faunal variability near the sediment water interface. The remaining length of
the core was sectioned into 2 cm intervals. The sediments from each interval were then washed with DI water through a 2mm
sieve, and the residue collected. Macrofauna and meiofauna in each section were identified with the aid of optical light
microscopy, and were preserved in an ethanol-glycol mixture (80% ethanol).

### 2.5 Organic Carbon Content

Dried porosity samples were ground by mortar and pestle and this homogenized sediment was used for $C_{org}$, $^{210}Pb$ and $^{137}Cs$
analyses. A portion of the ground sediment from the porosity sample was weighed (10-150 mg) and was placed into a 10 mL
exetainer tube and acidified with 10% phosphoric acid. The evolved gas was analyzed for Total $CO_2$ concentration using a
Picarro CRDS following procedures developed at USC (Subhas et al., 2015, 2017). This provided a measure of acid-reactive
C, assumed to equal C bound as $CaCO_3$. Another split of powder was weighed into tin capsules and combusted at 800° C on a
Costech CN analyzer with the $CO_2$ and $\delta^{13}C$ concentration also determined via the Picarro. USGS standards were used to
calibrate wt. % Total C in samples. The difference between total C and $CaCO_3$ carbon was taken as the % $C_{org}$. Replicates
indicate analytical uncertainties in this measurement of ±0.2 wt. % $C_{org}$ on samples that have 2-6 wt. % $C_{org}$.

### 2.6 Photographs and X-radiographs

Replicate cores from each multicorer sampling were photographed at University of California Los Angeles (UCLA). Cores
returned to the University of Southern California (USC) and were stored for 2-4 months to air-dry, which allowed the sediment
to lose water and consolidate. A router was used to remove a section of plastic core liner on opposite sides of the core tube.
The core was split into two halves with smooth cut faces from top to bottom using a wire. One split core was transferred to a
plastic tray with approximately a 2cm lip along the long edges. The wire was run along the top of the lip, yielding a uniform 2
cm thick slab of sample. Each slab was placed on a large sheet of Kodak film and x-rayed for 90-180 sec at 8 milliamps and
96 volts. Negatives were developed in a dark room.

### 2.7 Excess $^{210}Pb$ and $^{137}Cs$

Approximately 0.5- 1.0 g of dried, homogenized sediment was placed in 5 mL polypropylene test tubes for analysis by gamma
spectroscopy. Excess $^{210}Pb$ and $^{137}Cs$ activities in sediments were measured using high purity intrinsic germanium well-type



detectors (HPGe ORTEC, 120 cm$^3$ active volume). Detector efficiencies were determined by counting the activities of known standards in the same geometry as the samples. Standards used included IAEA-385 marine sediments, EPA Diluted Pitchblende SRM-DP2, and NIST $^{210}$Pb liquid solution. Samples were counted for 2-4 days, and the spectra (keV) were analyzed for the following radioisotopes: $^{210}$Pb (46), $^{214}$Pb (295), $^{214}$Pb (352), $^{214}$Bi (609), and $^{137}$Cs (661). The $^{226}$Ra activity (termed the supported $^{210}$Pb) was measured by counting the activity of the short lived $^{222}$Rn daughters ($^{214}$Pb and $^{214}$Bi). A

small 10% correction was applied to each sample to account for radon leakage, based on measurements of radon loss from similar sediments (Hammond et al., 1990). Excess $^{210}$Pb was determined by subtracting the supported $^{210}$Pb ($^{226}$Ra, Fig. A-1) from total $^{210}$Pb activity and correcting for decay between collection and analysis.

**2.8 $^{210}$Pb Calibration**


To verify the $^{210}$Pb results for SMB sediments by gamma spectroscopy, several samples from MUC 9 were also analyzed by isotope dilution alpha spectroscopy following procedures described by Huh et al. (1987). Briefly, 0.5 grams of sediment was placed in Teflon beakers and spiked with a known amount of $^{209}$Po spike (USC #65D). The $^{209}$Po spike was calibrated against a NIST certified $^{210}$Pb solution (NIST #4337). The sediments, along with the $^{209}$Po, were subject to a series of acid digestions

with HCl, HNO$_3$, HF, and HClO$_4$ or H$_2$O$_2$. After organics and silicates were dissolved, the solution was brought up in 50 mL 1N HCl, and ascorbic acid was added to complex iron. A 12-mm silver disk was placed on the bottom of the Teflon beaker, and the solution was stirred and heated at 90 ˚C for 3-5 hours to plate polonium. After 3-5 hours of plating, the silver disk was removed from the solution, washed with DIW, and dried with acetone. The silver disk was then measured using alpha spectrometry. The plated disks were placed 10 mm from Surface Silicon Barrier Detectors (ORTEC, 300 mm, average

efficiency = 18%) to measure the polonium isotopes. Backgrounds were counted regularly on each detector to monitor daughter product build-up. A background count was subtracted from each sample, and equation 2 was used to calculate the $^{210}$Pb in the dissolved sample.

$$A_{210_{Pb}} = \frac{^{210}N}{^{209}N} e^{(\lambda^{210}_{Po}\Delta t)} A_{209_{Po}} \tag{2}$$


where $^{210}$N and $^{209}$N are the background-corrected net counts of $^{210}$Po and $^{209}$Po; $\Delta t$ is the time that elapsed from midpoint of plating and midpoint of counting; A$_{209Po}$ is the activity of $^{209}$Po spike added in dpm; and $\lambda$ is the decay constant for $^{210}$Po. We assumed that $^{210}$Pb and $^{210}$Po (t$_{1/2}$ = 138 d) are in equilibrium for SMB sediments. A comparison of these two counting methodologies is shown in Figure 3 and provides convincing evidence that gamma and alpha spectroscopy yield identical

results.



### 2.9 Radiocarbon

Radiocarbon values were measured using the accelerator mass spectrometry (AMS) at the University of California Irvine
(UCI) Keck Carbon Cycle Accelerator Mass Spectrometry (KCCAMS) laboratory. Samples were subjected to HCl vapor for
four hours to acidify calcium carbonate, dried on a vacuum line, combusted, graphitized and then counted on the AMS. Sample
preparation backgrounds were subtracted, based on measurements of acidified glycine, ANU and Lycine. Radiocarbon results
have been corrected for isotopic fractionation according to the conventions of Stuiver and Polach (1977), with $\delta^{13}C_{org}$ measured
using a Costech ECS 4010 Analyzer - Delta V Plus IRMS at the University of California Riverside (UCR). The isotopic ratio
is given in delta notation relative to Vienna Pee Dee Belemnite (VPDB) for $\delta^{13}C$ values. Glycene, peach, acetate and house
soil were used as reference material, standard error ($1\sigma$) was <0.10‰.

### 3 Results

Sediment porosity declined with depth, with generally higher values in cores collected at deeper stations (Fig. 4). At all sites,
there was typically a porosity difference of ~0.2 between the sediment-water interface (SWI) and 30 cm depth horizon.
However, several cores showed notable interruptions in the monotonic decline in porosity with depth. Core MUC 9 and MUC
10 had intervals with lower porosities compared to the overall depth trend. Similar porosity anomalies were observed below
25 cm in MUC 9 and at 13-15 cm, ~22, and ~28 cm in MUC 10.

Only three cores (of those collected at depths > 320 m) had macrofauna obtained from sediment sectioning and sieving (Table
1). Notable was the abundance of sponge spicule clusters found throughout much of core MUC 8. An intact annelid worm
was found on the surface of MUC 11 (745 m), which had oxygen concentrations < 8 μM near the seafloor.

Weight percent $C_{org}$ content of the upper cm of the cores collected in 2016 showed a distinct trend of increasing %$C_{org}$ with
depth (Fig. 5). Basin sediments (MUC's 9 and 10) had 5-6 wt. % $C_{org}$ whereas slope sediments ranged from 2-5 wt. %. Cores
collected in the 1970's and 1980's show the same trend for core top %$C_{org}$ vs. water depth as the MUC cores (Gorsline, 1992;
Fig. 5).

Photographs of MUC cores showed light reddish-brown colored sediment near the surface of each core and a progression in
MUC 9 and 10 toward darker colored sediment with depth (Fig. 6). Only MUC 9 (907 m) had laminations visible by eye. The
sediment in the upper 10 cm from other cores (MUCs 10, 3, and 11) appeared homogeneous. MUC 11 showed a living
polychaete worm present at the sediment-water interface.



X-radiographs of MUC 9 and MUC 10 revealed distinct laminations (Fig. 7). MUC 9 showed clear sediment laminations down to approx. 15 cm. However, MUC 10, collected from a site only 14 m shallower did not show fine lamination, but broader banding was apparent down to 12 cm. Both cores showed zones of higher density material (light-colored in x-radiograph negative). A distinct higher density zone is seen in MUC 9 below 25 cm. Three zones of dense material were detected in the MUC 10 x-radiograph; the first was between 12-17 cm, the second at ~22 cm, and the third below 28 cm. These zones of
higher density material correspond with the zones of anomalously low porosity (Fig. 4).

### 3.1 Excess $^{210}$Pb and $^{137}$Cs

Values of excess $^{210}$Pb in surface sediments varied from 25 dpm/g at the shallow water sites to 100 dpm/g in deeper waters near the mid-basin (Fig. 8). Many of the cores from the shallower sites (<800 m) showed a constant activity of excess $^{210}$Pb in the top 3-5 cm, below which activity decreased exponentially. MUC 8 deviated from this trend and showed an increase in excess $^{210}$Pb at 9 cm. MUC 9 and MUC 10, which are the two cores in the inner basin collected from water depths greater than 850 meters, showed high values of excess $^{210}$Pb near the surface and an exponential decrease below the sediment-water
interface. Excess $^{210}$Pb in these two cores was restricted to the top 8 cm, whereas excess $^{210}$Pb penetrated deeper into the sediment of cores from the basin slope (MUC's 5, 6, 7, 8, and 3).

$^{137}$Cs profiles of MUC 9 and MUC 10 showed peaks between 4.5 and 2.5 cm depth, respectively (Fig. 9), whereas $^{137}$Cs profiles of cores taken along the slope showed very low values with large uncertainties.


### 3.2 Radiocarbon and $\delta^{13}C_{org}$

The organic carbon from selected intervals from MUC 9 and MUC 10 was measured for radiocarbon content and $\delta^{13}C_{org}$ to depths of 25 centimeters (Fig. 10, Fig. 11). $\Delta^{14}C$ (BP)* and $\delta^{13}C_{org}$ values were plotted vs. integrated mass to provide a
normalization to the porosity changes that occur downcore. $\Delta^{14}C$ (BP)* indicates a conventional radiocarbon age that was determined using the method of Stuiver and Polach (1977). A reservoir age adjustment was not applied to the $\Delta^{14}C$ (BP)* values. Data from 2 to 6 mass units (g/cm$^2$) indicates a linear relation of age and integrated mass (Fig. 10), consistent with an assumption that reservoir age and mass accumulation rate at these sites remained constant through this interval. In both cores, these intervals were fit with a regression to determine mass accumulation rate for the studied time period (depth ranges 7-16
cm in MUC9 and 7-14 cm in MUC10). Calculations of radiocarbon sedimentation rates for MUC 9 and 10 yield values of 9.0 and 12.0 mg/cm$^2$-yr, respectively. Both cores show a much younger value of $\Delta^{14}C$ in the upper 1 cm of sediments relative to the profile below this depth, likely reflecting bomb $^{14}C$ contamination. Below the zone that was fitted, $\Delta^{14}C$ (BP)* values for MUC 10 were quite erratic, due to several turbidites that were noted in this core. These turbidites also affected the $\delta^{13}C_{org}$





profiles (Fig. 11), introducing carbon that was isotopically lighter than the material immediately above and below. A larger
change in both isotopes was observed in MUC10 compared to MUC9.

## 4 Discussion

### 4.1 Excess [210]Pb as a measure of sedimentation rate

[210]Pb has proven to be a useful tracer for sediment accumulation rates in the Santa Monica Basin (Bruland et al., 1974; Huh et
al., 1989; Christensen et al., 1993) and similar environments (Souza et al., 2012) during the last 100 years. Past studies derived
mass accumulation rates (MAR) rates using [210]Pb by assuming a constant sedimentary flux of [210]Pb over the time scale
concerned (~100 years), negligible bioturbation, and strong absorption of [210]Pb to particles. These assumptions should be valid
in the deepest parts of the SMB where sediments are minimally disturbed by bioturbation as shown by laminations. An
exponential equation can be fitted to the excess [210]Pb versus integrated mass profiles, with integrated mass accounting for
compaction and small differences in porosity profiles between cores:

$$C = C_0 e^{-aI} \hspace{4cm} (3)$$

where C is excess [210]Pb in dpm/g, I is integrated mass at the midpoint of each sample interval, and a is the inverse scale length
($cm^2\ g^{-1}$) of the exponential curve. a is related to mass accumulation rate by:

$$a = \frac{\lambda}{s} \hspace{4cm} (4)$$

where $\lambda$ is the decay constant of [210]Pb ($0.0311\ yr^{-1}$) and s is mass accumulation rate in units of sediment mass/area-time.

Table 2 shows a compendium of mass accumulation rates for the central portion of SMB, obtained from cores collected during
a 42-year interval from 1974 to 2016, including our samples. All cores collected from depths >900 m showed MARs that were
remarkably consistent, averaging 17.5±2.1 $mg/cm^2$-yr. There was also no noticeable trend in MAR or variation in the amount
of excess [210]Pb at the sediment-water interface over time. Additionally, excess [210]Pb profiles were similar in structure
downcore. All cores in Fig. 12, except for those obtained in the present study, were retrieved by box corers, which can disturb
the top few centimeters of sediments (Huh et al., 1989). Yet all the cores collected from the deep basin showed remarkable
consistency, with no evidence of sedimentation rate change between the 1970's and 2016, as well as no evidence of core
disturbance.





We also compared [210]Pb profiles in cores retrieved from water depths 870-900 m (Fig. 13) to those collected from deeper sites. These cores showed the same MAR as the deeper sites although with more variation evidenced in the larger standard deviation (17.2± 5.5 mg/cm²-yr). However, we observed no systematic change as a function of year collected. Much of the variability in

MAR was driven by CaBS X BC2, which was collected at 870 m. Core CaBS V BC8 had a clear [210]Pb minimum in the upper 10 cm and featured a 'typical' [210]Pb profile only below this depth. The minimum and the offset of the extrapolated fit for the deeper points from the surface values suggest rapid input of material with low excess [210]Pb, most likely from a localized turbidite in this core. The eight cores collected from 870-900 meters showed similar values of surface excess [210]Pb as cores collected from sites >900 m. Although cores from the shallower depth range averaged the same MAR as the deeper cores, the

quality of the linear fit of excess [210]Pb versus integrated mass, as demonstrated by the average $R^2$ value, was poorer for cores 870-900 m (average $R^2 = 0.90$) compared to cores collected at depths of >900 m (average $R^2 = 0.99$ ), suggesting either episodic input of sediment with varying excess [210]Pb or possibly minor episodic disturbances.

**4.2 Changes in the areal extent of laminated sediments**


Christensen et al. (1993) and Huh et al. (1989) documented the concentric areal expansion of laminated sediments throughout the floor of SMB starting about 400 years B.P.  The onset of anoxia began in the south-east center basin, where the basin is deepest (> 900 m) and moved outward, asymmetrically, but in all directions (Fig. 14).  Using the presence of fine laminations as a proxy for oxygen deficiency and establishing the onset of lamination by assignment of age, a 'lateral' anoxic spreading

rate of 50-80 m yr[-1] was calculated.  Depending on the direction of a transect, the rate of anoxia spreading in vertical space varied considerably, from 0.06 m/year up the eastern slope to 0.19 m/year moving in an NNW direction (Fig. 14). This asymmetry may be attributed to the major circulation pattern of deep basin water, in which waters from the San Pedro Basin enter SMB from the south-east and travel counter-clockwise. In such a flow, the eastern slope of the SMB would be bathed by overlying waters with slightly more oxygen than waters on the north-north-west side of the basin.  The overall expansion of

anoxic waters may reflect both a reduction of oxygen in waters entering the basin, as well as increased oxygen consumption within deep basin waters. The latter could arise from either an increase rain-rate of labile carbon, or a reduction in water replacement rates.

Only two of the 2016 cores analyzed in the present study showed sedimentary layering in x-radiographs (MUC 9 and MUC

10).  For the deepest core, MUC 9, there was clear evidence of finely-laminated sediments in the top 15 cm (Fig. 7).  MUC 10, which is located in the southern SMB, near the connection to San Pedro Basin, showed a banding of sediments in the upper few cm of the core (1-2 cm width) but no fine lamination, suggesting minimal bioturbation.  Given MUC 10's location in relation to the spread of oxygen deficiency, the absence of finely-laminated sediments in the upper few cm suggests that the spread of oxygen deficiency has not extended to this location.  Furthermore, MUC 3 (777 m), which is right at the boundary

of the zone of oxygen deficiency defined by Christensen et al. (1993) had no indications of laminations, and [210]Pb clearly





showed a mixed zone in the upper 4 cm (Fig. 8). These two MUC cores make it tempting to suggest the oxygen deficiency zone is contracting; however, we can conclude with confidence that the position of the zone in SMB has not changed markedly since cores were last analyzed in the 1980's.

**4.3 Changes in mass accumulation rates—A comparison of [210]Pb and [14]C methods**

Interpretation of [210]Pb and [14]C profiles in terms of sediment accumulation rate rely on assumptions that the delivery of these radio-tracers have been consistent and continuous, and that the sediment has not been disturbed via mixing (typically bioturbation). The assumption of consistency is generally assumed to be true in basins that receive sediments via hemipelagic
sedimentation, and the assumption of no-disturbance is supported by sediment fabric as revealed by x-radiography (Fig. 7).

The similarity of [210]Pb profiles in core MUC 9, which shows fine lamination structure in the top 8 cm, and core MUC 10, which shows coarser sediment banding, is evidence that some minor disturbance in the latter core may have obscured lamination structure, but disturbance has been insufficient to change the [210]Pb profile. Both of these cores yield similar sediment
accumulation rates, ~17 mg/cm$^2$-yr, and no evidence for a change in sedimentation rate over the lifetime of [210]Pb, which is approximately 100 years. Because the Bruland (1974) core does not show any evidence of a change in sedimentation rate through the life of [210]Pb, we can conclude that sedimentation has been constant in SMB since around the late 1800's. We find it striking that sediment accumulation offshore from an urban center has remained constant, even though the region has grown from a small town to the present 15+ million-person megalopolis of Los Angeles.

While accumulation rates remained constant during the period of rapid population growth in Los Angeles, the [14]C accumulation rates, not including those horizons that lie within turbidite deposits, define a sedimentation rate for the period ~ about 1700-1900 C.E. that is less than that defined by [210]Pb. In both MUCs 9 and 10 cores, [14]C dated sediment horizons yield sediment accumulation rates of 9-12 mg/cm$^2$-yr compared to 17.5 ± 2.1 derived from [210]Pb profiles (Table 2). This trend of increasing
sedimentation toward the recent is opposite of what might have been predicted due to the trapping of sediment via flood-control engineering of the LA River. However, our data are consistent with the proposal made by Tomašových and Kidwell (2017) noting that sometime in the mid-late 1800s sediment delivery to the coastal zone of the SMB increased. Tomašových and Kidwell (2017) based their interpretation on the change in the SMB shelf ecosystem structure that occurred at that time. From the loss of a filter-feeding ecosystem from the SMB shelf environments, Tomašových and Kidwell (2017) infer an
increase in fine sediment delivery to the SMB shelf.

The determinations of [14]C sediment accumulation rates could be biased or incorrect if there has been a changing input of particulate organic matter (PO[14]C) to the SMB. In MUC 10 there is an obvious section of core where [14]C age dates are old and $\delta^{13}C_{org}$ values are light, relative to the trend defined by the other data. However, these two measurements are from a turbidite



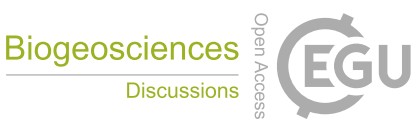

deposit (Fig. 11) and are consistent with the interpretation that such a deposit contains older, terrestrially derived (perhaps more refractory) POC (Meyers, 1994). MUC9 may also show a minor influence from this turbidite, but the effect is subtle.  A plot of $\delta^{13}C_{org}$ versus integrated mass of MUC 9 and MUC 10 show a trend to slightly lighter $\delta^{13}C_{org}$ near the top, although the change is very small.  (Fig 11). The $^{14}$C profiles for both cores appear to show slightly older sediments than expected, just above the 2 mass unit horizon, suggesting a change to additional input of older carbon associated with the modest change in

$\delta^{13}C_{org}$. The changing trend could record a terrestrial source, but the data are not clear-cut. While there may have been a change in the source of carbon (and sediment) in the late 1800s, the data prior to this time indicates there has been a step-function change in sediment accumulation rate that took place sometime around the late 1800's.

It is tempting to suggest that reservoir age, instead of sedimentation rate, could explain the offset of $^{14}$C values down core.

However, if the sedimentation rate determined from the excess $^{210}$Pb profile at MUC 9 is assumed constant down core to a depth represented by 2-6 mass units, then 236 years would have elapsed during this interval (4 g/cm$^2$/0.017 g/cm$^2$-yr=236 yr). If the $^{210}$Pb MAR applies through this interval, and the $^{14}$C values record changes in reservoir age and not sedimentation rate, the reservoir age for organic carbon would need to increase by 160 years, at a steady rate, over the time period represented by 2-6 mass units.  While this cannot be dismissed, it would imply a higher upwelling rate in the past, and there seems no evidence

for this.

Another explanation of the $^{14}$C MAR not agreeing with the $^{210}$Pb MAR is that there was a higher proportion of old terrestrial carbon reaching the sediments during the past.  However, the lack of a significant change in $\delta^{13}C_{org}$ through this interval makes this process an unlikely explanation. We think it most likely that an increase in MAR occurred somewhere in the late 1800's

and speculate that further $^{14}$C analysis of laminated sediments, preserved under low oxygen conditions, is the best way to support this conclusion.

A previous study that considered Holocene sediment accumulation in SMB (Romans et al., 2009) found that the hemipelagic sediment accumulation rate for the late Holocene averaged ~10 mg/cm$^2$-yr (this rate determined from their linear sediment

accumulation rate and the extrapolation of our porosity data to a depth of 2 m), although the turbidite accumulation rate was substantially greater. This is a value consistent with the MAR we found from the $^{14}$C dated section of our cores, only 150-300 years before present. Thus, it appears that hemi-pelagic sedimentation in SMB has been very consistent over the past 1000's of years and has increased by ~70% in the recent 100-150 years.

**4.4 Biological Activity in Low Oxygen Environments**

Only three cores analyzed for this study had macrofauna present, these were MUC 12 (508 m), MUC 8 (695 m) and MUC 11 (745 m). All three cores were collected from bottom waters with <20 µM oxygen concentration and the deeper two sites have




<10 µM oxygen. The living annelid found in MUC 11 is evidence that macrofauna can be active and hence potentially act to
bioturbate at low oxygen levels (<5 µM).

A preponderance of sponge spicules was found in replicate cores from the location of MUC 8. This is also a site bathed in
waters with <5 µM oxygen. In addition to the core sectioned for biological inspection, a core that was x-radiographed shows
the presence of a partially articulated demosponge within the sediment column at ~8 cm depth (Morine, 2017). These sponges
are not infaunal, thus the most plausible explanation for the high spicule abundance in these cores is that this sediment zone
has been populated by sponges for >100 years.

Prior to the work of Christensen et al. (1993), Malouta et al. (1981) mapped out the area of bioturbation throughout the SMB
using x-radiographs of basin cores. Using disturbances in laminated sediments as a proxy for different levels of bioturbation,
3 different zones were assigned: completely disturbed laminae, partially disturbed laminae, and fine laminae present.
Completely disturbed laminae were cores that showed no laminations or banding and were usually found on the shelf and
slopes of the SMB, typically shallower than 750m. Partially disturbed laminae were characterized by some hints of banding
and suggested minimal bioturbation. Lastly, finely laminated sediments were zones of no bioturbation and were located in the
deep, central basin at depths greater than 900m. The areas to which Malouta et al. (1981) assigned these zones of bioturbation
correlate with our cores obtained in 2016, suggesting minimal changes in organism activity vs. depth during the last 40 years.
Additionally, our work shows that laminae can be largely obscured and yet a [210]Pb profile from a slightly bioturbated core
(MUC 10) can appear nearly indistinguishable from a profile from a well-laminated core (MUC 9).

**5 Conclusions**


A suite of cores was collected in 2016 to explore whether changes in the areal extent of laminated sediments and their mass
accumulation rates have changed during recent decades. Only one core analyzed in 2016 showed finely laminated sediments
in x-radiographs (MUC 9 at 907 m). Other cores showed cm-scale layering of sediments or no layering at all. The absence of
finely laminated sediments in MUC 10 (893 m) and MUC 3 (777 m) suggest that the rate of oxygen deficiency-spreading, as
noted by Huh et al. (1989) and Christensen et al. (1993) has not increased remarkably since cores were last collected in the
1980's. It is possible that the rate of anoxic bottom water spreading has declined or even possibly reversed with a slightly
shrinking area of laminated sediments. X-radiographs of laminations from cores collected in this study were compared to the
different levels of bioturbation mapped out 40 years ago in the SMB. The zones of bioturbation correlate with cores collected
in 2016, suggesting minimal change in macrofaunal activity (assumed a proxy for bottom oxygen concentrations) during the
last 40 years.

Through a summary of previously published profiles and new measurements of $^{210}$Pb in sediment cores from this study, a comparison of mass accumulation rate records in the central portion of SMB was examined in cores collected over a 42 year span. Mass accumulation rates for the deepest parts of the SMB basin (>900m) have been remarkably consistent since the late

1800s, averaging $17.5 \pm 2.1$ mg/cm$^2$-yr. At slightly shallower sites (870-900m), accumulation rates showed a little more variability, but yield the same accumulation rate, averaging $17.5 \pm 5.5$ mg/cm$^2$-yr. Excess $^{210}$Pb near the sediment-water interface was also consistent for all cores deeper than 870 m during the last 4 decades. The consistency of sedimentation rates, both for the past 40 years but also for the lifetime of $^{210}$Pb, ~100 years, is remarkable given the changes that have occurred in the Los Angeles region over the past century.


$\Delta^{14}$C values below 7 cm depths suggest sediment accumulation rates were lower prior to the late 1800s. MUC 9 and MUC 10 reveal sedimentation rates of 8.6 and 12.0 mg/cm$^2$-yr prior to the late 1800s, which is 55-75% of the rates determined for younger sediments from the excess $^{210}$Pb profiles. The slower sedimentation rate persisted back into the late Holocene (Roman et al., 2009). The increase in MAR appears to be a step-function change. A possible explanation, offered by Tomašových and

Kidwell (2017), is that sedimentation increased between 1850-1900 due to the rapid rise of cattle grazing and increased erosion. Why this increased rate remained high after urban development and why it should have remained so constant are unanswered questions, particularly following installation of debris basins that trapped a large portion of the sediment flux. Perhaps these basins largely captured coarse debris, while the fine sediment fraction that contributes to hemi-pelagic input has not been captured, but its input has been augmented by cattle grazing and subsequent urban development.


Evidence of sedimentary change in the SMB during the last 40 years is astonishingly absent. Mass accumulation rates, laminated sediments, extent of bioturbation, and % C$_{org}$ have changed little during this time. The only parameter that appears to have clearly changed in the last 200 years is the sedimentation rate, which shows a step-function increase slightly before 1900.


## 6 Acknowledgments

The work was supported by a NOAA Sea Grant (USC, award# NA14OAR4170089) Award to William Berelson and Tina Treude. The Petroleum Research Fund of the American Chemical Society provided funding to Timothy Lyons and Maria

Figueroa. Tina Treude was further supported by a faculty research grant from the University of California Los Angeles.




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





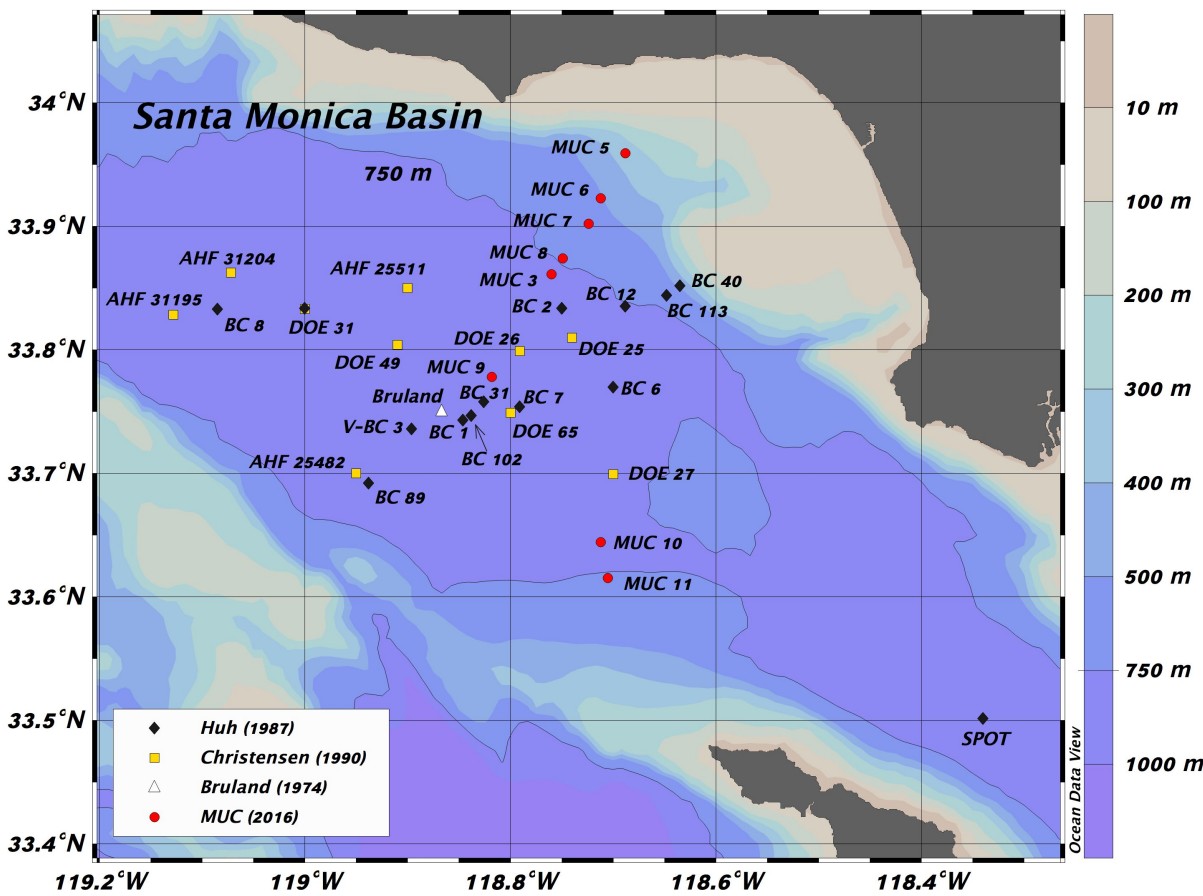

**Figure 1: All cores and coring locations presented in this paper. Acronyms 'MUC', 'BC', and 'AHF' indicate Multi-Core, Box Core, and Allan Hancock Foundation (see Table 2 for more details on coring locations).**





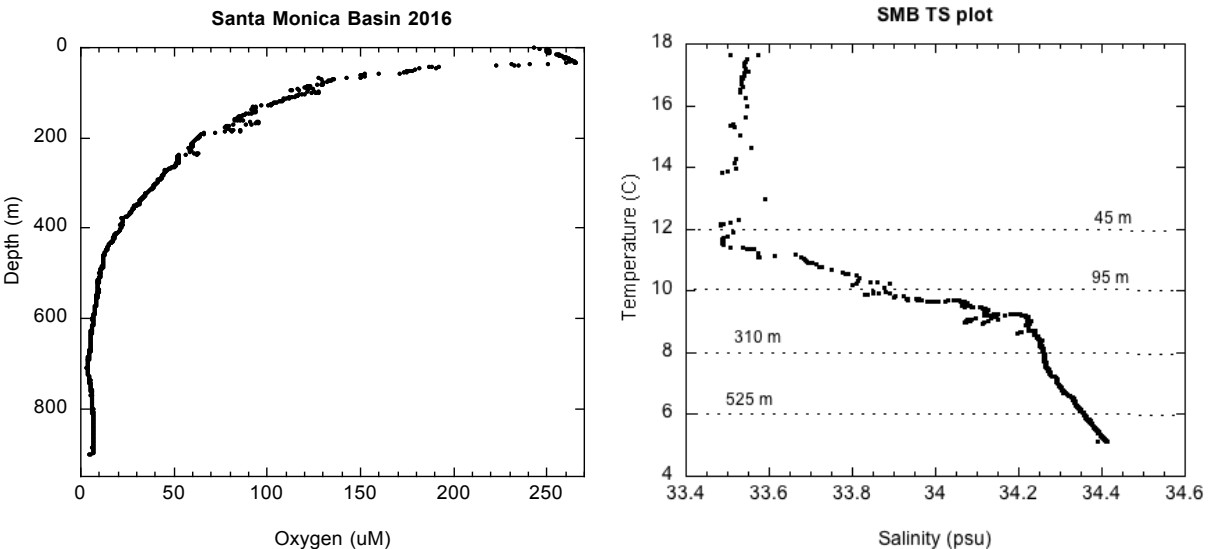

**Figure 2: Oxygen and T-S plot for SMB obtained in spring 2016 (MUC 9).**


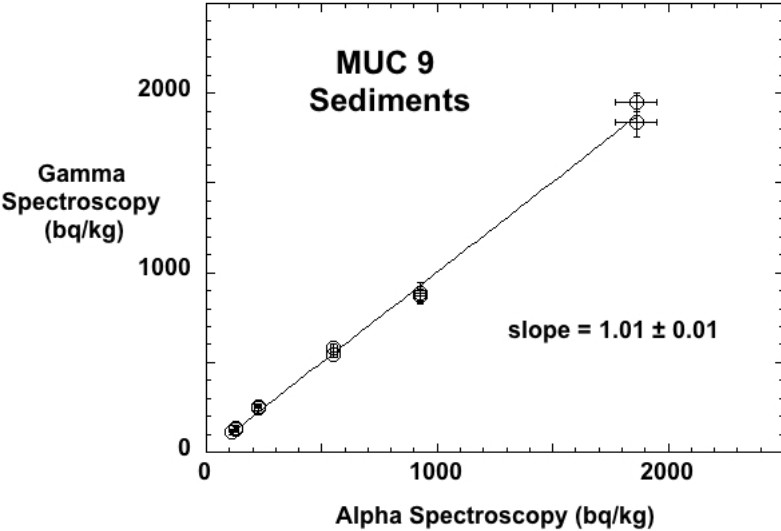

**Figure 3: Specific activities of [210]Pb determined from gamma and alpha spectroscopy for Santa Monica Basin sediments (MUC-9). The linear regression is forced through zero. All activities are corrected for salt content.**





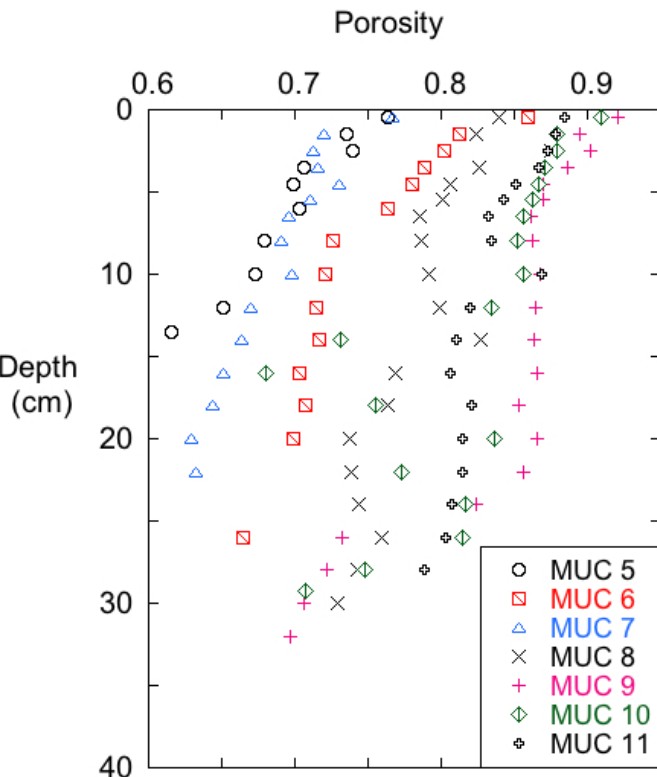


**Figure 4: Porosity profiles for SMB 2016 MUC cores.**



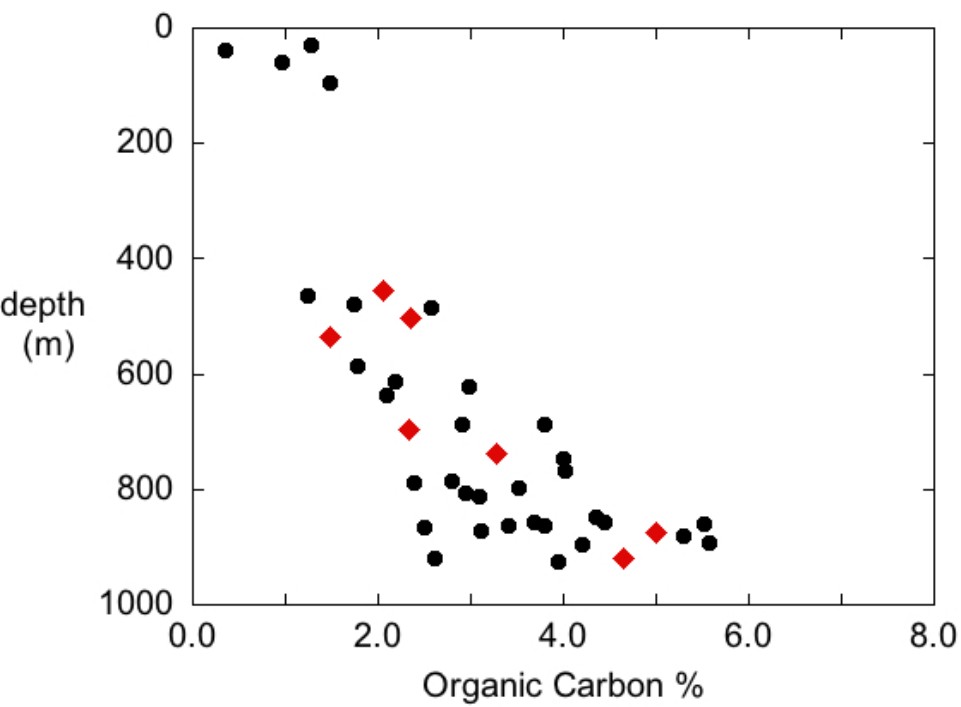

**Figure 5: %C_org content for the 0-1 cm intervals from MUC cores (red triangles) and data from Gorsline (unpublished box core results). Box core data also represent the upper (0-2 cm) sediment C_org fraction.**




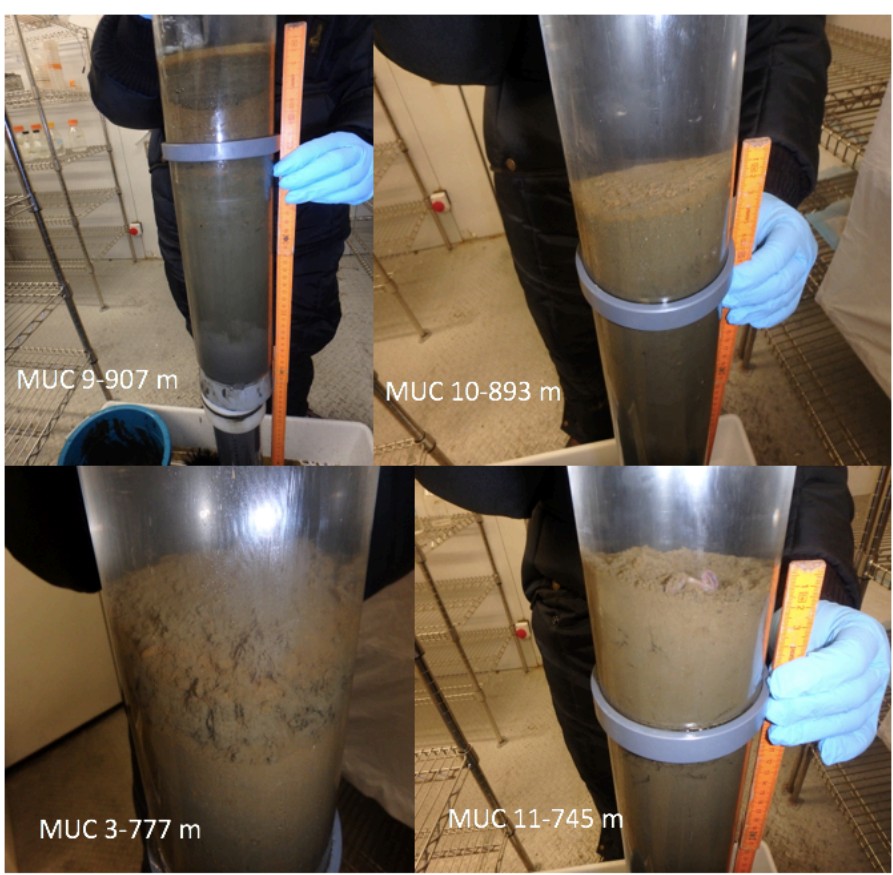

**Figure 6: Photographs of selected 2016 MUC cores.**






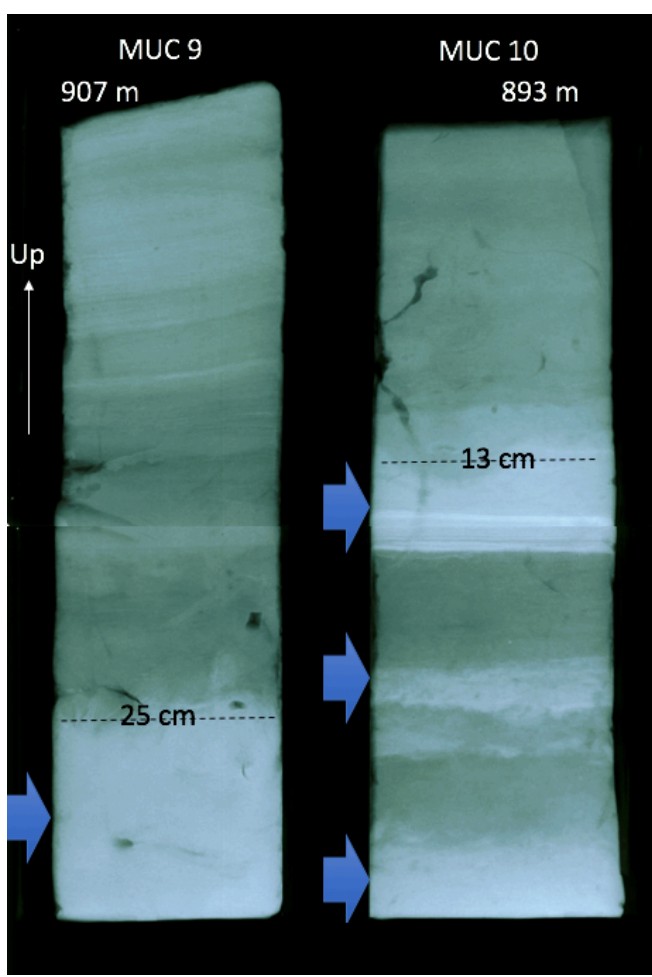

Figure 7: X-radiographs of cores MUC 9 and 10. Arrows designate location of turbidites which show up in x-ray as lighter colored (denser). Also note the fine laminations visible in MUC9.



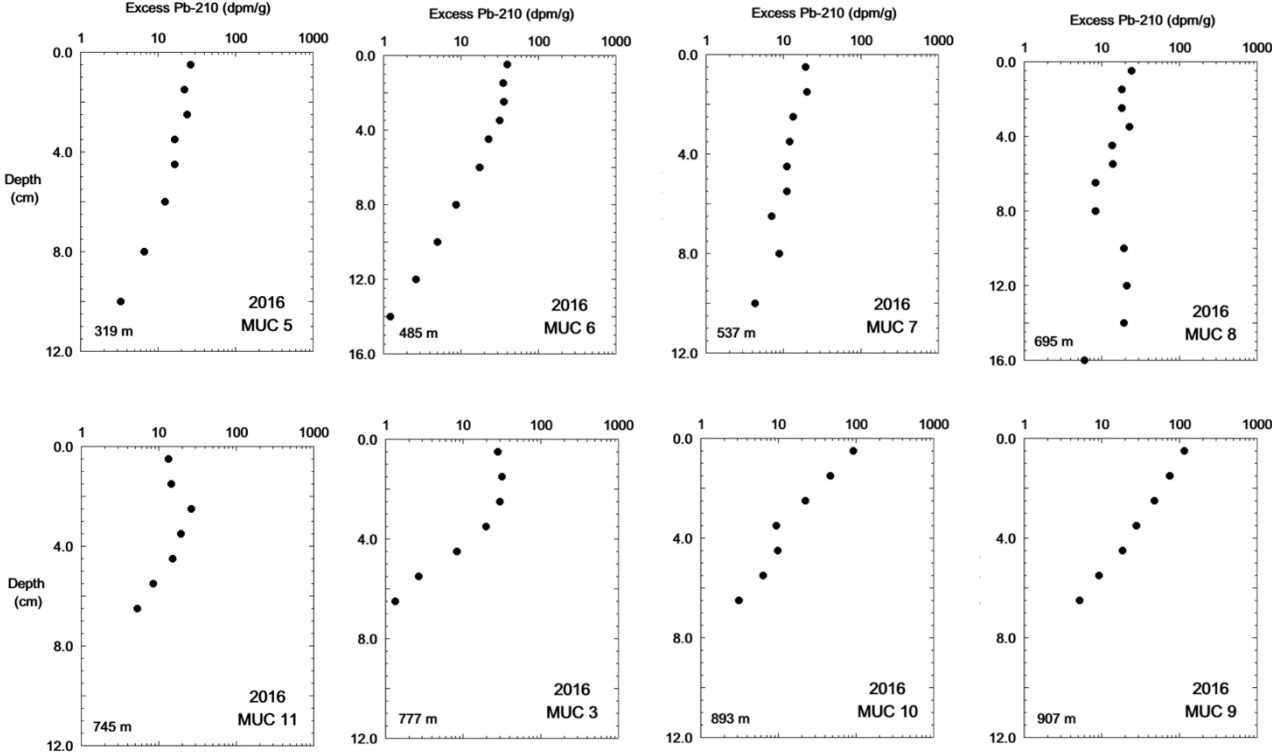

**Figure 8: Eight multi-cores sampled for** [210]**Pb in the Santa Monica Basin. The points are plotted in the middle of the depth interval (given in Table 2). Note that the depth scales are slightly different for different cores. Excess** [210]**Pb values below 5 dpm/g have large uncertainty (± 2 dpm/g) based on counting statistics.**





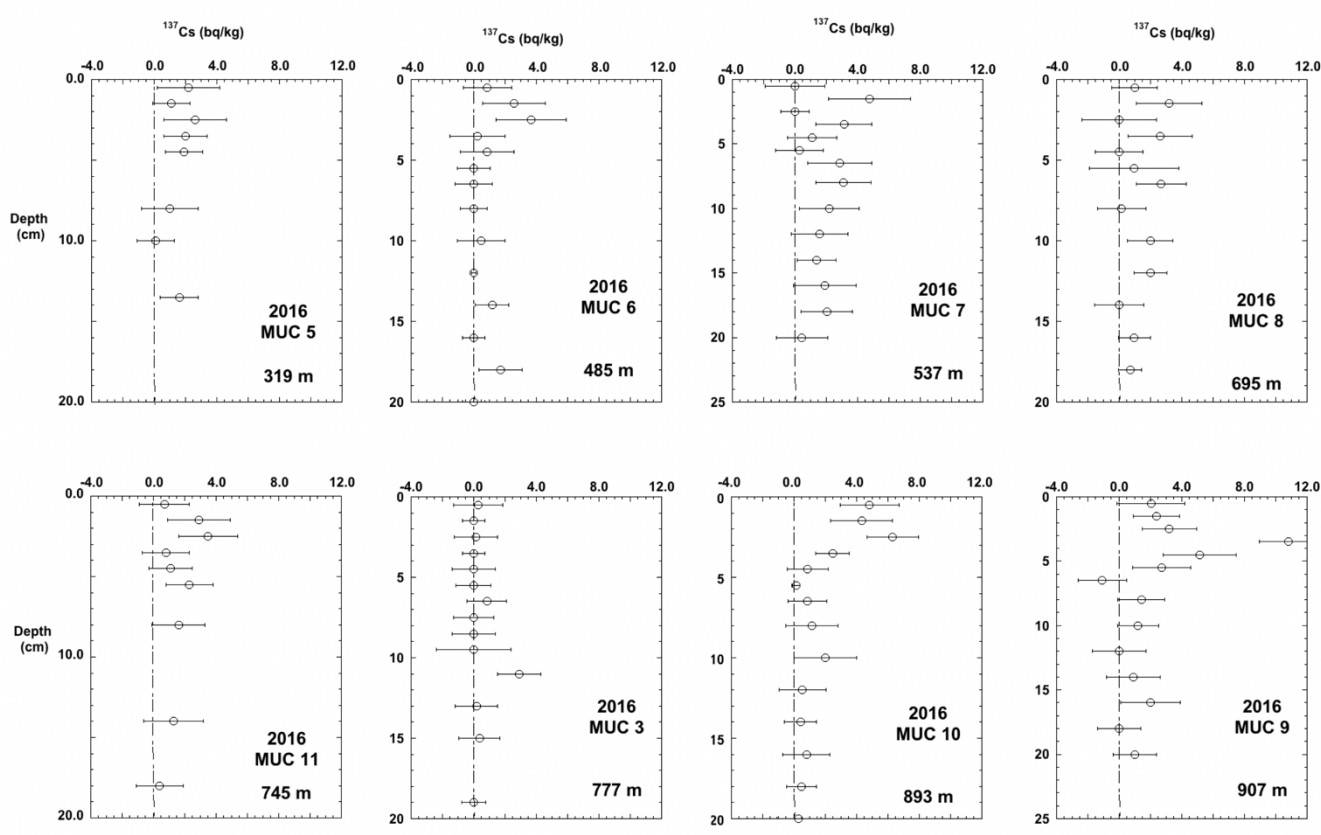


**Figure 9: Eight multi-cores sampled for $^{137}$Cs in the Santa Monica Basin. MUC 9 and MUC 10 were the only cores with a clear $^{137}$Cs peak. All other cores had no defined peak.**



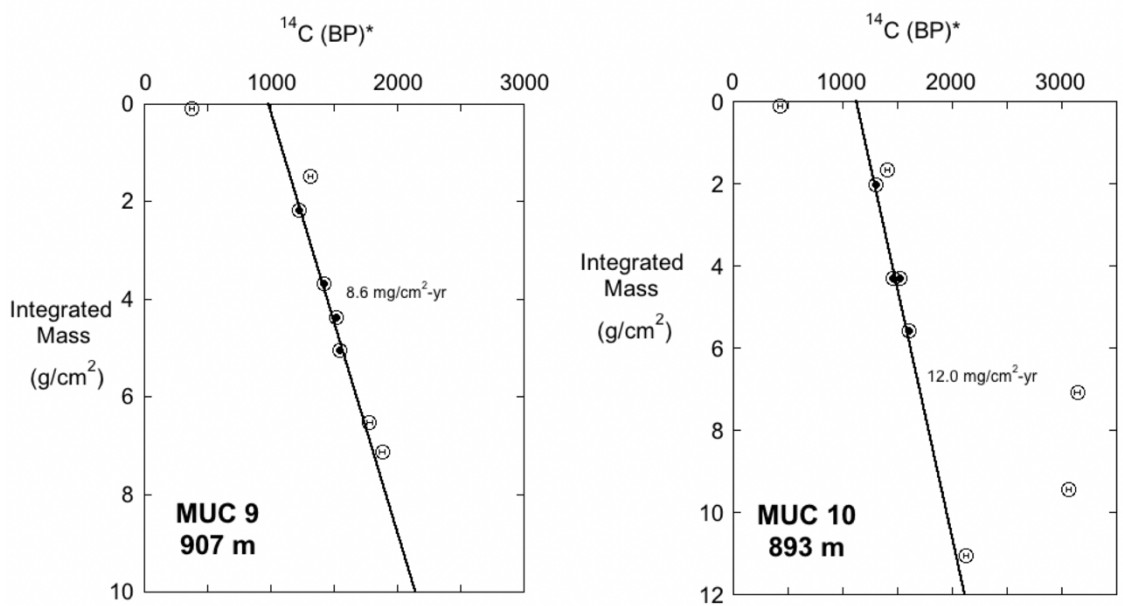

**Figure 10: $\Delta^{14}$C (BP)\* vs. Integrated Mass (g/cm$^2$) for SMB cores MUC 9 and MUC 10. $\Delta^{14}$C (BP)\* denotes conventional radiocarbon age without assigning a reservoir age. If reservoir age and mass accumulation rate are unchanging, the plots should be linear. A linear fit was applied to the solid circles (lying between 2 and 6 mass units, horizons 5-18 cm depth range). These solid circles should not be influenced by turbidites or "bomb" carbon. The regression slope defines mass accumulation in mg cm$^{-2}$ y$^{-1}$. The depth equivalent to 2 g/cm$^2$ represents an age of approximately 120 years B.P.**





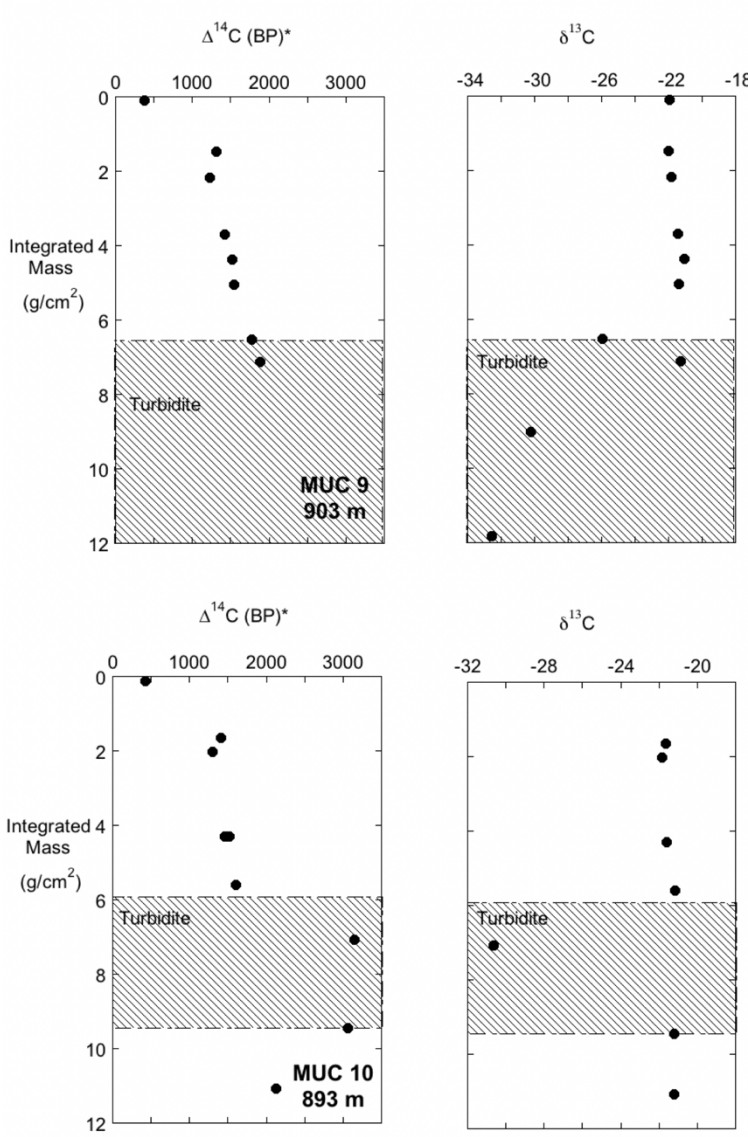

Figure 11: $\Delta^{14}C$ (BP)* vs. Integrated Mass (g/cm²) as in Fig. 10 except including designation of turbidite region (hachured) and corresponding d¹³C data.





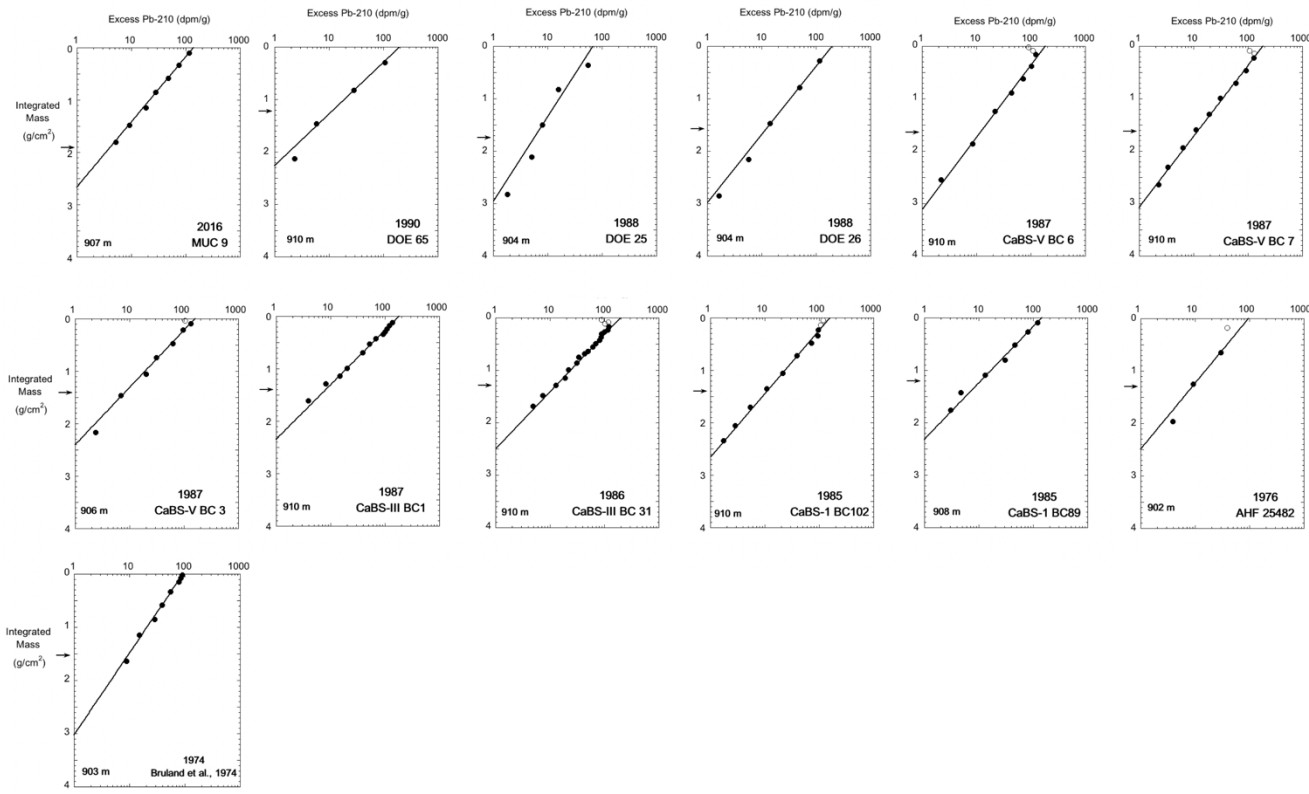


**Figure 12: Semi-log plot of Excess ²¹⁰Pb activity vs. integrated mass for thirteen cores sampled in the Santa Monica Basin between the years 1974-2016. All 13 cores are from depths greater than 900 meters. The linear fit to these plots yield slopes that define the mass accumulation rate (see Table 2). Arrows on the y-axis indicate the integrated mass equivalence to the year 1900 CE.**



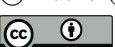



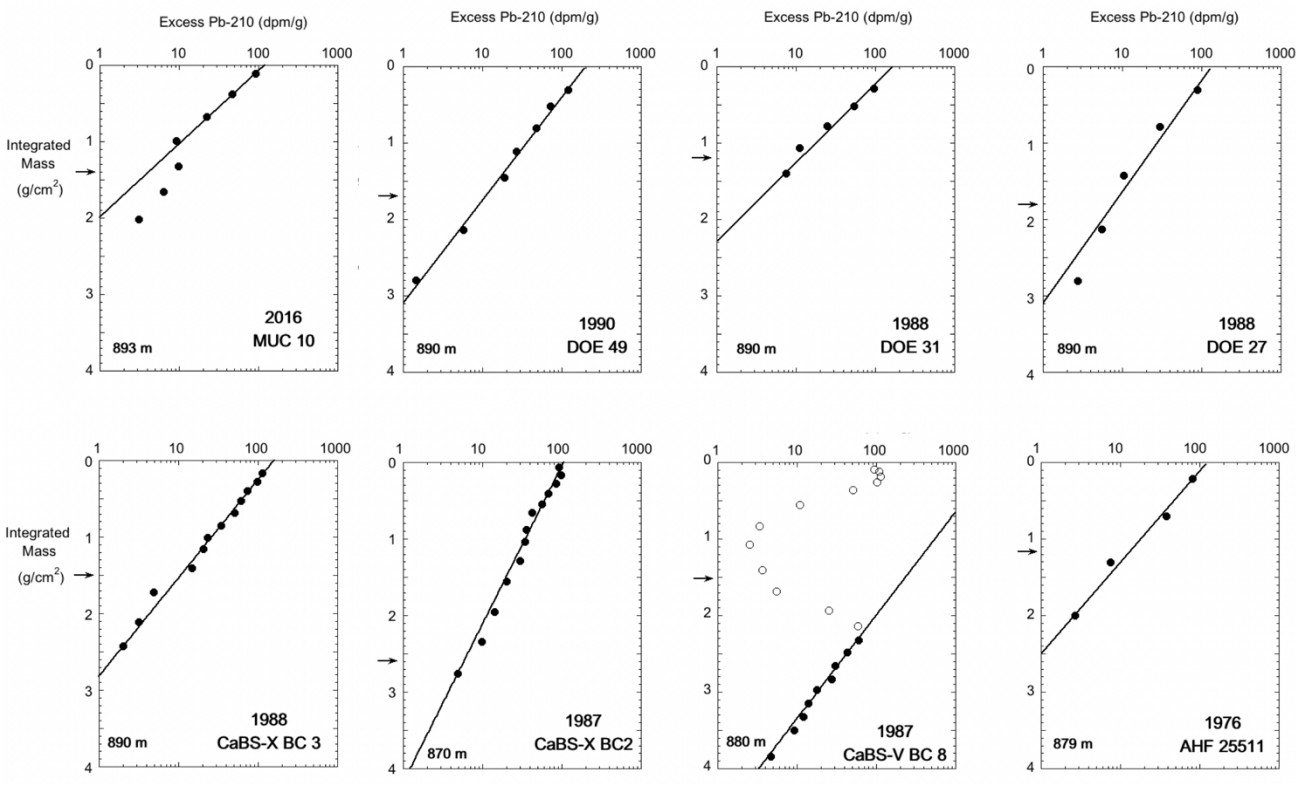

Figure 13: Same as Fig. 12 but for 8 cores obtained from depths between 870-900 meters.






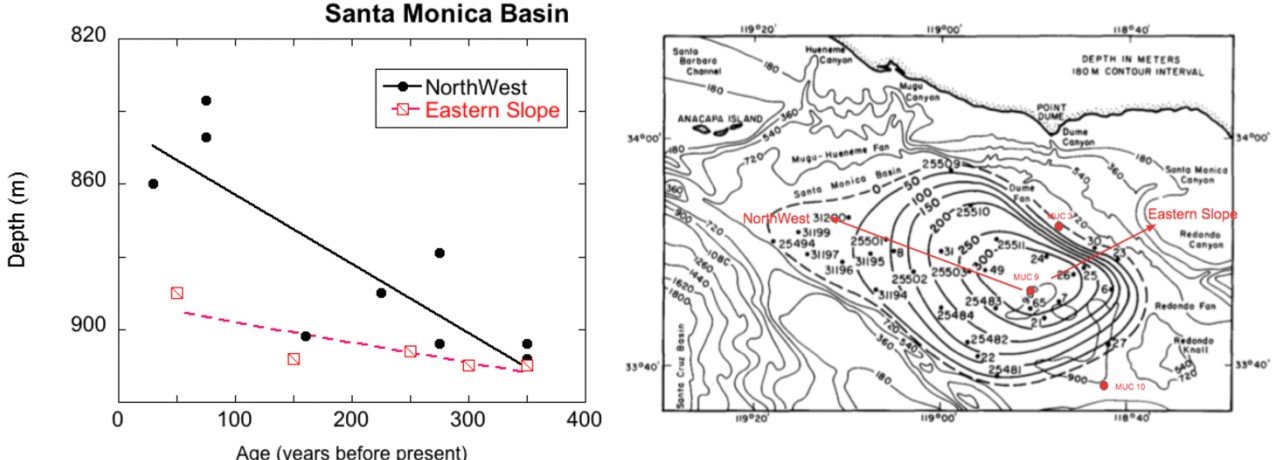

**Figure 14: Spreading of the laminated sediments area, defined as a change in depth over time for two transects. Left panel shows**
**the spreading rate of laminated sediments as they progress upslope (shallower depths) moving toward the recent. The figure on the**
**right is modified from Christensen et al. (1994) based on his time scale and shows the growth in areal extent of laminated sediments**
**since 300 years ago. The location of three MUC cores obtained in 2016 are also shown. The expansion of laminated sediment**
**accumulation has occurred more rapidly in the NW direction than in the Eastern transect.**


**Table 1: Macrofauna for selected SMB 2016 MUC cores.**

| Bottom Depth | Core ID | Core Interval | Description |
|---|---|---|---|
| m | | cm | |
| 508 | MUC 12 | 5-6 | Annelid, Polychaete, Arenicola sp. |
| | | 9-11 | Porifera |
| 695 | MUC 8 | 0-1 | Porifera, Demosponge-partially articulated |
| | | 11-31 | Porifera, Demosponge-abundant spicules |
| 745 | MUC 11 | | Annelid, Polychaete, Arenicola sp. |






**Table 2: Station ID, year collected, mass flux, depth, inventory and excess 210Pb at sediment water interface (SWI) for all cores greater than 800 meters depth in the Santa Monica Basin. The first 13 cores are from deeper than 900 meters, and the last 8 cores are from 800-900 meter water depths. 210Pb Inventory was also computed but values are not discussed in this article (\* indicates a graphical integration was used, others are from fitting parameters). References: [1] = this work, [2] = Christensen et al., 1991, [3] = Huh et al., 1989; [4] is Bruland, 1974.**

| Year Collected | Station ID | Mass Flux mg/cm$^2$-yr | Depth m | Excess $^{210}$Pb @ SWI dpm/g | Inventory dpm/cm$^2$ | Reference |
|---|---|---|---|---|---|---|
| 2016 | MUC-9 | 16.8 | 907 | 140 | 71 | [1] |
| 1990 | DOE 65 | 13.6 | 910 | 200 | 90 | [2] |
| 1988 | DOE 25 | 20.8 | 904 | 70 | 59 | [2] |
| 1988 | DOE 26 | 17.7 | 904 | 190 | 110* | [2] |
| 1987 | CaBS V BC6 | 18.8 | 910 | 163 | 114* | [3] |
| 1987 | CaBS V BC7 | 18.5 | 910 | 190 | 111* | [3] |
| 1987 | CaBS V BC3 | 15.8 | 906 | 160 | 76* | [3] |
| 1986 | CaBS III BC31 | 14.9 | 910 | 174 | 88* | [3] |
| 1986 | CaBS III BC 1 | 15.8 | 910 | 182 | 82* | [3] |
| 1985 | CaBS I BC102 | 16.6 | 910 | 159 | 92 | [3] |
| 1985 | CaBS I BC89 | 14.1 | 908 | 145 | 100 | [3] |
| 1976 | AHF 25842 | 17.8 | 902 | 97 | 53* | [2] |
| 1974 | Bruland, 1974 | 20.7 | 903 | 94 | 69 | [4] |
| **Average (±SSD)** | | **17.5 ± 2.1** | | | | |
| | | | | | | |
| 2016 | MUC-10 | 12.2 | 893 | 120 | 54 | [1] |
| 1990 | DOE 49 | 19.1 | 890 | 200 | 111* | [2] |
| 1988 | DOE 31 | 13.3 | 890 | 130 | 78* | [2] |
| 1988 | DOE 27 | 20.1 | 860 | 120 | 90 | [2] |
| 1988 | CaBS X BC3 | 17.0 | 890 | 160 | 89* | [3] |
| 1988 | CaBS X BC2 | 29.3 | 870 | 107 | 96* | [3] |
| 1987 | CaBS V BC 8 | 16.8 | 880 | N/A | N/A | [3] |
| 1976 | AHF 25511 | 15.1 | 879 | 120 | 68* | [2] |
| **Average (±SSD)** | | **17.2 ± 5.5** | | | | |






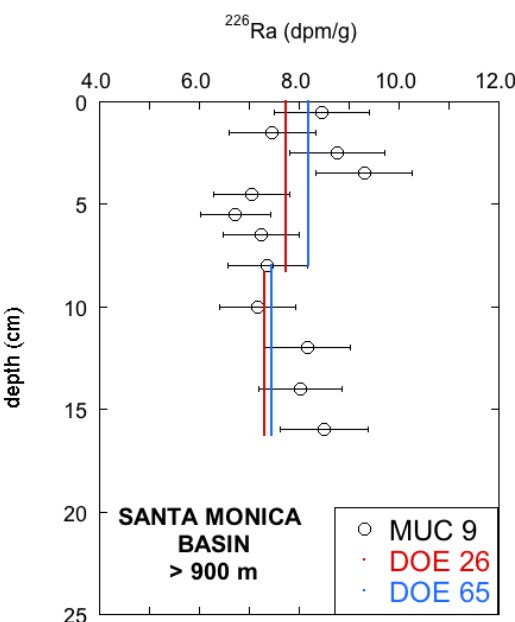

**Figure A-1:**[226]**Ra values for SMB cores MUC 9, DOE 65, and DOE 25. All three cores are sampled greater than 900 meters depth and within close approximation to each other. MUC 9 was collected in 2016, and the measurement was based on gamma spectroscopy corrected for 10% Rn loss. The other two cores were collected in the late 1980's and a composite sample was analysed by Rn ingrowth from the dissolved sediment (Christensen et al., 1993). All** [226]**Ra values.**
