# Peer review of "Evidence of Changes in Sedimentation Rate and Sediment Fabric in a Low Oxygen Setting: Santa Monica Basin, CA"

_Biogeosciences, 2019_

## Referee Comment (RC1) · Christopher Fuller (Referee) · 27 Dec 2019

This paper seeks to combine results of recent cores with those collected over the past 45 years to assess changes in sediment accumulation rate and spreading of suboxic conditions to shallower depths in Santa Monica Basin in response to urbanization. 210Pb derived sediment mass accumulation rates (MAR) are combined with presence/absence of laminations or infauna. The overall all conclusion of little change in both mass accumulation rate and extent of the low oxygen condition are generally supported by the new data in conjunction with a summary of previous studies. After addressing comments below, this paper will be a useful contribution to further the

understanding of changes in sediment and geochemical dynamics in this near-shore environment.

How are constant activities in the upper 3-5 cm of cores from shallower depths defined (lines 245-250)? Are the activities are within uncertainty of each other? A factor of two decrease is shown in the upper 5 cm in some of these profiles (MUC 5, 6, 7) compared to deeper depths, so not "constant" but instead upper 5 cm has a different slope than below, which can be interpreted as higher accumulation rate and/or mixing. This warrants further discussion such as whether there is a increase in MAR, or mixing is the likely cause. The reasons for excluding cores from discussion needs to be made clearer such as in lines 329-339.

Turbidite layers are noted in core MUC10 (line 267), which could impact 14C profiles. Were these layers accounted for in deriving rates? Figure 10 and 11 would benefit from showing depth as well as mass on y axis.

Section 4.1. It is unclear if mass accumulation rates from 210Pb profiles of the previous studies were re-determined here or if rates from previous papers are accepted as is. Did the earlier efforts account for sediment compaction?

The comparison of rates within the depth regimes (Table 2 and section 4.1) uses the mean of all cores within a depth group. The means have a small standard error. However, the range in rates is a factor of 1.7 so that stating that rates are "consistent" is somewhat misleading. It would be more instructive to determine the uncertainty in each mass accumulation rate from the uncertainty in slope of unsupported 210Pb versus cumulative mass, then evaluate if rates among a depth regime are significantly different.

It would be helpful in section 4.1 to state (or remind the reader) the basis for dividing the core sites into >900 and <900-meter water depth groups.

Section 4.2, lines 318-320. It is unclear how the assignment of age was made to establish the onset of laminations, and the resulting spreading rate. Are these estimates from the literature or derived here? In either case, this warrants additional explanation.

The inferred step-wise change in mass accumulation rates in section 4.3 is based on 14C profiles from two cores but the inference seems erroneous. Lines 346-350 state similar MAR of 17 mg/cm2/yr for cores MUC 9 and 10, yet Table 2 lists rates of 16.8 and 12.2, respectively, for MUC 9 and 10. In addition, the comparison of two 14C rates from cores MUC 9 and 10 is made to the 210Pb MAR averages of all cores in Table 2, not to the MAR for specific two cores. Instead, the MUC10 C14 rate of 12 is in very good agreement with its 210Pb rate of 12.2 (per Table 2). Something seems a miss here in concluding a step-wise change for both sites. The ensuing discussion on lines 380-395 needs to be revised accordingly.

The statement of "nearly indistinguishable" 210Pb profiles on line 420 doesn't follow the difference in 210Pb derived MAR in Table 2 for these two cores.

Statement on lines 441-443 of consistent surface 210Pb activity is not supported by the range of almost a factor of 3 shown in Table 2. Revise accordingly.

---

## Short Comment (SC1) · 29 Dec 2019

This manuscript is extremely useful for understanding the history of sedimentation and bottom-water conditions in the Santa Monica Basin. Authors show x-ray radiographs from MUC9 and MUC10 in figure 7 and do not show x-rays from other stations, probably because other cores were not showing laminations. However, differences in the degree of bioturbation or some indistinct banding can be maybe visible even in x-rays of those cores – as suggested at lines 415-420. Therefore, it can be useful to add x-rays of all stations to assess spatial and bathymetric variation in bioturbation.

One another information about the finding that sediment accumulation rates was slower

prior to 1900 ́CE — this change was also recently documented on the basis of a decline in abundance of a deposit-feeding bivalve Nuculana taphria in two sediment cores collected on the Palos Verdes and San Pedro shelves - this sand-dwelling species, one of the most important contributors to molluscan community on the shelf prior to the 20th century, significantly declined in abundance sometimes at the 19th/20th century transition or during the 19th century at mid-shelf water depths (Fig. 11 in Tomasovych et al. 2019 in Paleoceanography and Paleoclimatology) (clearly prior to the major onset of wastewater pollution, $\sim$ in 60-70s of the 20th century).

Reference Tomasovych, A., Kidwell, S.M., Alexander, C.R. and Kaufman, D.S., 2019. Paleoceanography and Paleoclimatology 34, 954-977

---

## Referee Comment (RC2) · Anonymous Referee #2 · 8 Feb 2020

This paper utilizes sediment cores collected over the past 45 years to determine changes in sediment accumulation rates in Santa Monica Basin in response to urbanization using 14C and 210Pb methodologies. The overall conclusion shows that the mass accumulation rate did not show evidence of significant changes over this period. The paper will be a somewhat useful contribution with minor changes

Specific comments: 1. The authors should clearly identify which 210Pb data were measured and which rates are from previously published work. 2. The Pb-210 method section is long and can be summarized by references appropriate publications, given that 210Pb is a commonly used method. 3. The figure for alpha vs gamma calibration

for Pb-210 can be moved to supplement and is not directly relevant, especially since some of the co-authors have long established history of working in these isotopes. 4. Pb-210 should explicitly state this method is based on constant input and constant sedimentation rate (e.g. Appleby; Cochran papers). 5. The constant rate of sedimentation can be partly verified by looking at the goodness of fit and any apparent break in slope. In this context it will be more appropriate to plot Fig xx as ln(Pbex) vs depth and provide the regression equation and r2. 6. The mass accumulation rates calculated using the slope of regression has an associated uncertainty term based on fit, which should be translated to the uncertainty term for the determined sedimentation rates. Since change in sedimentation rate is an important objective of this work, the uncertainty associated with determined sedimentation rate can give a sense of how much it could have changed. 7. On the same note it might be worthwhile to do a sensitivity study for the 210Pb model used, to determine its ability to capture subtle changes in sedimentation rate. A single sedimentation rate is determined by linear regression of downcore distribution of 210Pb excess, where it is assumed each data point provides equally precise information about the deterministic part of the total process variation. However the 210Pb excess activities in deeper layers are lower with larger errors compared to shallower depths. Thus it is possible, barring major shift in sedimentation rate, less dramatic changes in sedimentation rates may not be detectable.

---

## Author Comment (AC1) · 27 Feb 2020

Christopher Fuller (Referee) ccfuller@usgs.gov

This paper seeks to combine results of recent cores with those collected over the past 45 years to assess changes in sediment accumulation rate and spreading of suboxic conditions to shallower depths in Santa Monica Basin in response to urbanization.
210Pb derived sediment mass accumulation rates (MAR) are combined with presence/absence of laminations or infauna. The overall all conclusion of little change in both mass accumulation rate and extent of the low oxygen condition are generally supported by the new data in conjunction with a summary of previous studies. After addressing comments below, this paper will be a useful contribution to further the understanding of changes in sediment and geochemical dynamics in this near-shore environment.

How are constant activities in the upper 3-5 cm of cores from shallower depths defined (lines 245-250)? Are the activities within uncertainty of each other? A factor of two decrease is shown in the upper 5 cm in some of these profiles (MUC 5, 6, 7) compared to deeper depths, so not "constant" but instead upper 5 cm has a different slope than below, which can be interpreted as higher accumulation rate and/or mixing. This warrants further discussion such as whether there is a increase in MAR, or mixing is the likely cause. The reasons for excluding cores from discussion needs to be made clearer such as in lines 329-339.

Answer: This was re-written and explained as constant activities in the upper 2-3 cm of cores. 3-5 cm was a little too much as most cores showed an exponential decrease below this horizon. We believe these cores on the shelf are disturbed by bioturbation and are not representing an increase in accumulation rates. We concluded this because previous studies have clearly demonstrated bioturbation via excess 234Th. While this study did not measure 234Th, it can be safely assumed that most cores were influenced by bioturbation because x-radiography showed no laminations (mixing has smeared the laminations). I will reiterate this in the text.

Turbidite layers are noted in core MUC10 (line 267), which could impact 14C profiles. Were these layers accounted for in deriving rates? Figure 10 and 11 would benefit from showing depth as well as mass on y axis.

Answer: Yes, turbidite layers were accounted for in determining MAR. This is now more

clearly stated in Figure 10 and discussed in section 4.3.

Section 4.1. It is unclear if mass accumulation rates from 210Pb profiles of the previous studies were re-determined here or if rates from previous papers are accepted as is. Did the earlier efforts account for sediment compaction?

Answer: This has been made clearer with my new statements in section 4.1. MAR were taken as is from previous studies (but I also re-calculated them myself and obtained similar results).

The comparison of rates within the depth regimes (Table 2 and section 4.1) uses the mean of all cores within a depth group. The means have a small standard error. However, the range in rates is a factor of 1.7 so that stating that rates are "consistent" is somewhat misleading. It would be more instructive to determine the uncertainty in each mass accumulation rate from the uncertainty in slope of unsupported 210Pb versus cumulative mass, then evaluate if rates among a depth regime are significantly different.

Answer: To address this point, I added MAR errors from the slope for each station/core. Also, I added standard deviation of the mean in each depth range (STDEV/SQRT(N)).

It would be helpful in section 4.1 to state (or remind the reader) the basis for dividing the core sites into >900 and <900-meter water depth groups.

Answer: I added a comment to this effect in this section.

Section 4.2, lines 318-320. It is unclear how the assignment of age was made to establish the onset of laminations, and the resulting spreading rate. Are these estimates from the literature or derived here? In either case, this warrants additional explanation.

Answer: The estimates were from the previous studies. I stated in clearer terms how they determined the previous rates.

The inferred step-wise change in mass accumulation rates in section 4.3 is based on

14C profiles from two cores but the inference seems erroneous. Lines 346-350 state similar MAR of 17 mg/cm2/yr for cores MUC 9 and 10, yet Table 2 lists rates of 16.8 and 12.2, respectively, for MUC 9 and 10. In addition, the comparison of two 14C rates from cores MUC 9 and 10 is made to the 210Pb MAR averages of all cores in Table 2, not to the MAR for specific two cores. Instead, the MUC10 C14 rate of 12 is in very good agreement with its 210Pb rate of 12.2 (per Table 2). Something seems a miss here in concluding a step-wise change for both sites. The ensuing discussion on lines 380-395 needs to be revised accordingly.

Answer: There was a typo of MUC10's MAR: it should be 14.1 mg/cm2-yr and not 12 mg/cm2-yr. However, what was being said in this section is that MAR derived from 14C are lower (9-12 mg/cm2-yr) prior to 1900 CE than "all" the MARs derived from 210Pb values over the last 40 years, which averages 17 mg/cm2-yr. Our inference of 'step-wise change' comes from evaluating a rate of 9-12 and comparing that to a rate of 17±2. I know we have only two cores to prove that MARs were slower prior to 1900 CE, but I think the confusion lies in that MUC10 has a slightly lower MAR than the average of the rest of the cores. However, given that only 1 or 2 cores from the 18 cores sampled over the last 40 years shows a MAR similar to the 2 profiles of 14C, suggests that an increase in sedimentation most likely has occurred in the last 150 years.

The statement of "nearly indistinguishable" 210Pb profiles on line 420 doesn't follow the difference in 210Pb derived MAR in Table 2 for these two cores.

Answer: Deleted this sentence

Statement on lines 441-443 of consistent surface 210Pb activity is not supported by the range of almost a factor of 3 shown in Table 2. Revise accordingly.

Answer: The reviewer is correct, 3 cores had a factor of 2 lower integrated activity than the rest of cores which averaged around 170 dpm/g. Two of those 3 cores were taken in the 1970's and the upper 1 cm was most likely disturbed due to box cores used, thus

lowering its activity. A sentence to this effect has been added to the text.

---

## Author Comment (AC2) · 27 Feb 2020

Anonymous (Referee)

This paper utilizes sediment cores collected over the past 45 years to determine changes in sediment accumulation rates in Santa Monica Basin in response to urbanization using 14C and 210Pb methodologies. The overall conclusion shows that the mass accumulation rate did not show evidence of significant changes over this period. The paper will be a somewhat useful contribution with minor changes

[Figure]

Specific comments: 1. The authors should clearly identify which 210Pb data were measured and which rates are from previously published work.

Answer: This has been addressed in section 4.1 Excess 210Pb as a measure of sedimentation rate, by stating clearly where each accumulation rates were derived.

2. The Pb-210 method section is long and can be summarized by references appropriate publications, given that 210Pb is a commonly used method.

Answer: I am assuming this is referring to the first paragraph in section 4.1, Excess 210Pb as a measure of sedimentation rate, where this section discusses the method and shows 2 equations that were used to determine sedimentation rates via 210Pb. I have removed the 2 equations, shortened the paragraph, and stated the appropriate references for the 210Pb method.

3. The figure for alpha vs gamma calibration for Pb-210 can be moved to supplement and is not directly relevant, especially since some of the co-authors have long established history of working in these isotopes.

Answer: As per this reviewer's suggestion, the section, 2.8 210Pb Calibration, was moved to supplement section of this paper.

4. Pb-210 should explicitly state this method is based on constant input and constant sedimentation rate (e.g. Appleby; Cochran papers).

Answer: We now explicitly say this in section 4.1, Excess 210Pb as a measure of sedimentation rate: constant initial concentration model is what we use.

5. The constant rate of sedimentation can be partly verified by looking at the goodness of fit and any apparent break in slope. In this context it will be more appropriate to plot Fig xx as ln(Pbex) vs depth and provide the regression equation and r2.

Answer: I believe this is shown in Figure 12 and 13. While I do not have R2 or regression equation on each plot, I do have, in Table 2, each plot's accumulation rate and its

**[BGD]**

Interactive
comment

associated uncertainty.

6. The mass accumulation rates calculated using the slope of regression has an associated uncertainty term based on fit, which should be translated to the uncertainty term for the determined sedimentation rates. Since change in sedimentation rate is an important objective of this work, the uncertainty associated with determined sedimentation rate can give a sense of how much it could have changed.

Answer: Uncertainties in each mass accumulation rate has been added to Table 2 by determining the uncertainty for each slope regression.

7. On the same note it might be worthwhile to do a sensitivity study for the 210Pb model used, to determine its ability to capture subtle changes in sedimentation rate. A single sedimentation rate is determined by linear regression of downcore distribution of 210Pb excess, where it is assumed each data point provides equally precise information about the deterministic part of the total process variation. However, the 210Pb excess activities in deeper layers are lower with larger errors compared to shallower depths. Thus, it is possible, barring major shift in sedimentation rate, less dramatic changes in sedimentation rates may not be detectable.

Answer: A sensitivity calculation assuming a step-change reduction of 40% in accumulation rate in 1930 (2 half-lives before the Bruland et al., (1974) core) shows 210Pb has marginal sensitivity to resolving the timing of the change (computed profile not shown).

---

## Author Comment (AC3) · 27 Feb 2020

Christopher Fuller (Referee) ccfuller@usgs.gov

This paper seeks to combine results of recent cores with those collected over the past 45 years to assess changes in sediment accumulation rate and spreading of suboxic conditions to shallower depths in Santa Monica Basin in response to urbanization. 210Pb derived sediment mass accumulation rates (MAR) are combined with presence/absence of laminations or infauna. The overall all conclusion of little change in both mass accumulation rate and extent of the low oxygen condition are generally supported by the new data in conjunction with a summary of previous studies. After addressing comments below, this paper will be a useful contribution to further the understanding of changes in sediment and geochemical dynamics in this near-shore environment.

How are constant activities in the upper 3-5 cm of cores from shallower depths defined (lines 245-250)? Are the activities within uncertainty of each other? A factor of two decrease is shown in the upper 5 cm in some of these profiles (MUC 5, 6, 7) compared to deeper depths, so not "constant" but instead upper 5 cm has a different slope than below, which can be interpreted as higher accumulation rate and/or mixing. This warrants further discussion such as whether there is a increase in MAR, or mixing is the likely cause. The reasons for excluding cores from discussion needs to be made clearer such as in lines 329-339.

Author's Response: This was re-written and explained as constant activities in the upper 2-3 cm of cores. 3-5 cm was a little too much as most cores showed an exponential decrease below this horizon. We believe these cores on the shelf are disturbed by bioturbation and are not representing an increase in accumulation rates. We concluded this because previous studies have clearly demonstrated bioturbation via excess 234Th. While this study did not measure 234Th, it can be safely assumed that most cores were influenced by bioturbation because x-radiography showed no laminations (mixing has smeared the laminations). We will reiterate this in the text.

Turbidite layers are noted in core MUC10 (line 267), which could impact 14C profiles. Were these layers accounted for in deriving rates? Figure 10 and 11 would benefit from showing depth as well as mass on y axis. Author's Response: Yes, turbidite layers were accounted for in determining MAR. This is now more clearly stated in Figure 10 and discussed in section 4.3.

Section 4.1. It is unclear if mass accumulation rates from 210Pb profiles of the previous studies were re-determined here or if rates from previous papers are accepted as is. Did the earlier efforts account for sediment compaction? Author's Response: This has been clarified in section 4.1. MAR were taken as is from previous studies (but we also re-calculated them and obtained similar results).

The comparison of rates within the depth regimes (Table 2 and section 4.1) uses the mean of all cores within a depth group. The means have a small standard error. However, the range in rates is a factor of 1.7 so that stating that rates are "consistent" is somewhat misleading. It would be more instructive to determine the uncertainty in each mass accumulation rate from the uncertainty in slope of unsupported 210Pb versus cumulative mass, then evaluate if rates among a depth regime are significantly different. Author's Response: To address this point, we added MAR errors from the slope for each station/core.

We also added standard deviation of the mean in each depth range (STDEV/SQRT(N)).

It would be helpful in section 4.1 to state (or remind the reader) the basis for dividing the core sites into >900 and <900-meter water depth groups.

Author's Response: We added a comment to this effect in this section.

Section 4.2, lines 318-320. It is unclear how the assignment of age was made to establish the onset of laminations, and the resulting spreading rate. Are these estimates from the literature or derived here? In either case, this warrants additional explanation.

Author's Response: These estimates were from the previous studies. We stated in clearer terms how they determined the previous rates.

The inferred step-wise change in mass accumulation rates in section 4.3 is based on 14C profiles from two cores but the inference seems erroneous. Lines 346-350 state similar MAR of 17 mg/cm2/yr for cores MUC 9 and 10, yet Table 2 lists rates of 16.8 and 12.2, respectively, for MUC 9 and 10. In addition, the comparison of two 14C rates

**BGD**

from cores MUC 9 and 10 is made to the 210Pb MAR averages of all cores in Table 2, not to the MAR for specific two cores. Instead, the MUC10 C14 rate of 12 is in very good agreement with its 210Pb rate of 12.2 (per Table 2). Something seems a miss here in concluding a step-wise change for both sites. The ensuing discussion on lines 380-395 needs to be revised accordingly.

Author's Response: There was a typo of MUC10's MAR: it should be 14.1 mg/cm2-yr and not 12 mg/cm2-yr (we corrected accordingly). However, what was being said in this section is that MAR derived from 14C were lower (9-12 mg/cm2-yr) prior to 1900 CE than "all" the MARs derived from 210Pb values over the last 40 years, which averages 17 mg/cm2-yr. Our inference of 'step-wise change' comes from evaluating a rate of 9-12 and comparing that to a rate of 17±2. We are aware that we have only two cores to demonstrate that MARs were slower prior to 1900 CE, but we believe the confusion lies in that MUC10 has a slightly lower MAR than the average of the rest of the cores. However, given that only 1 or 2 cores from the 18 cores sampled over the last 40 years shows a MAR similar to the 2 profiles of 14C, suggests that an increase in sedimentation most likely has occurred in the last 150 years.

The statement of "nearly indistinguishable" 210Pb profiles on line 420 doesn't follow the difference in 210Pb derived MAR in Table 2 for these two cores. Author's Response: We deleted this sentence.

Statement on lines 441-443 of consistent surface 210Pb activity is not supported by the range of almost a factor of 3 shown in Table 2. Revise accordingly. Author's Response: The reviewer is correct, 3 cores had a factor of 2 lower integrated activity than the rest of cores which averaged around 170 dpm/g. Two of those 3 cores were taken in the 1970's and the upper 1 cm was most likely disturbed due to box cores used, thus lowering its activity. A sentence to this effect has been added to the text.

---

## Author Comment (AC5) · 11 Mar 2020

Anonymous (Referee)

This paper utilizes sediment cores collected over the past 45 years to determine changes in sediment accumulation rates in Santa Monica Basin in response to urbanization using 14C and 210Pb methodologies. The overall conclusion shows that the mass accumulation rate did not show evidence of significant changes over this period. The paper will be a somewhat useful contribution with minor changes Specific com-

ments: 1. The authors should clearly identify which 210Pb data were measured and which rates are from previously published work. Author's Response: This has been addressed in section 4.1 Excess 210Pb as a measure of sedimentation rate, by stating clearly where each accumulation rates were derived. 2. The Pb-210 method section is long and can be summarized by references appropriate publications, given that 210Pb is a commonly used method. Author's Response: We assume this comment is referring to the first paragraph in section 4.1, Excess 210Pb as a measure of sedimentation rate, where this section discusses the method and shows two equations that were used to determine sedimentation rates via 210Pb. We removed the two equations, shortened the paragraph, and stated the appropriate references for the 210Pb method.

3. The figure for alpha vs gamma calibration for Pb-210 can be moved to supplement and is not directly relevant, especially since some of the co-authors have long established history of working in these isotopes. Author's Response: As per this reviewer's suggestion, section,2.8 210Pb Calibration was moved to the supplement section of this paper. 4. Pb-210 should explicitly state this method is based on constant input and constant sedimentation rate (e.g. Appleby; Cochran papers). Author's Response: We now explicitly say this in section 4.1, Excess 210Pb as a measure of sedimentation rate: constant initial concentration model is what we use.

5. The constant rate of sedimentation can be partly verified by looking at the goodness of fit and any apparent break in slope. In this context it will be more appropriate to plot Fig xx as ln(Pbex) vs depth and provide the regression equation and r2. Author's Response: We do not show R2 or regression equation for each plot, but we do have (see Table 2) each plot's accumulation rate and its associated uncertainty. The associated uncertainty in each plots accumulation rate should be a suitable indicator for goodness of fit instead of R2 (all plots showed R2 value of 0.99 or higher). 6. The mass accumulation rates calculated using the slope of regression has an associated uncertainty term based on fit, which should be translated to the uncertainty term for the determined sedimentation rates. Since change in sedimentation rate is an important objective of

this work, the uncertainty associated with determined sedimentation rate can give a sense of how much it could have changed. Author's Response: Uncertainties in each mass accumulation rate has been added to Table 2 by determining the uncertainty for each slope regression.

7. On the same note it might be worthwhile to do a sensitivity study for the 210Pb model used, to determine its ability to capture subtle changes in sedimentation rate. A single sedimentation rate is determined by linear regression of downcore distribution of 210Pb excess, where it is assumed each data point provides equally precise information about the deterministic part of the total process variation. However, the 210Pb excess activities in deeper layers are lower with larger errors compared to shallower depths. Thus, it is possible, barring major shift in sedimentation rate, less dramatic changes in sedimentation rates may not be detectable. Author's Response: A sensitivity calculation assuming a step-change reduction of 40% in accumulation rate in 1930 (2 half-lives before the Bruland et al., (1974) core) shows 210Pb has marginal sensitivity to resolving the timing of the change (computed profile not shown).